# A Review: Construction of Statistical Distributions

**DOI:** 10.3390/e27121188

**Published:** 2025-11-23

**Authors:** Kai-Tai Fang, Yu-Xuan Lin, Yu-Hui Deng

**Affiliations:** 1Guangdong Provincial Key Laboratory of Interdisciplinary Research and Application for Data Science (IRADS), Beijing Normal-Hong Kong Baptist University, 2000 Jintong Road, Tangjiawan, Zhuhai 519087, China; ktfang@bnbu.edu.cn (K.-T.F.); yuxuanlin@bnbu.edu.cn (Y.-X.L.); 2The Key Lab of Random Complex Structures and Data Analysis, The Chinese Academy of Sciences, Beijing 100045, China; 3Faculty of Science and Technology, Beijing Normal-Hong Kong Baptist University, 2000 Jintong Road, Tangjiawan, Zhuhai 519087, China

**Keywords:** construction of statistical distribution, distribution family, entropy, function of random variables, mean square error, quasi-Monte Carlo, representative points, statistical distribution

## Abstract

Statistical modeling is fundamentally based on probability distributions, which can be discrete or continuous and univariate or multivariate. This review focuses on the methods used to construct these distributions, covering both traditional and newly developed approaches. We first examine classic distributions such as the normal, exponential, gamma, and beta for univariate data, and the multivariate normal, elliptical, and Dirichlet for multidimensional data. We then address how, in recent decades, the demand for more flexible modeling tools has led to the creation of complex meta-distributions built using copula theory.

## 1. Introduction

Statistical distributions form the foundation of probability theory, statistical inference, and predictive modeling. They provide a mathematical framework for describing the randomness, variability, and uncertainty inherent in data. Different types of data—univariate or multivariate, discrete or continuous—require different distributions, including mixture models for complex patterns.

Numerous books have offered comprehensive introductions to useful distributions. The seminal four-volume work by Johnson et al. [1,2,3,4] provides an exceptionally detailed encyclopedia of distributions and their interrelations, though its sheer scope can make it challenging for readers to locate specific information efficiently. Other key texts include Fang and Xu’s book [5] (in Chinese, which limits its accessibility), Fang et al. [6] focusing treatment on multivariate symmetric distributions, Johnson [7] reviewing statistical simulation for multivariate distributions, and the new work by Fang et al. [8] offering a comprehensive study on representative points of distributions and their applications.

The purpose of this review is to synthesize various methods for constructing statistical distributions. By presenting a unified overview of both traditional and modern techniques, we aim to equip readers with the knowledge to understand existing distributions and develop new ones to meet evolving data modeling needs.

This paper is structured as follows. Section 2 introduces construction methods for univariate distributions, encompassing both traditional and newer models. Section 3 briefly reviews methods for constructing discrete distributions, including both univariate and multivariate frameworks. Section 4.4 then details construction techniques for multivariate statistical distributions, specifically the families of elliptical distributions and meta-distributions created via copula methods. The paper concludes with final remarks in Section 5.

## 2. Construction of Univariate Statistical Distributions

This section introduces several approaches for constructing univariate continuous distributions:(1)Distribution families, including location-scale families, the Pearson system, and exponential families;(2)Functions of random variables via stochastic representations;(3)Parameter expansion generalizes existing families by adding parameters. For example, normal→ skew normal; exponential → Weibull.(4)Generalized families via transformation of the cdf. A powerful and prolific approach for generating new families of distributions is through the functional transformation of an existing baseline cdf, *F*(*x*). These transformations introduce new parameters, thereby creating more flexible distributions capable of modeling complex data behaviors.(5)Combination methods, such as mixture distributions, compound distributions, convolutions, and copula-based techniques.

### 2.1. Distribution Families

A distribution family is defined by a specific structural property. Among many such families, the following three are particularly well established in the literature.

#### 2.1.1. Location-Scale Distribution Family

A distribution *F*(*x*; *μ*, *σ*) belongs to the location-scale family if its probability density function (pdf) can be expressed as(1)$$f\left(x\right)=\frac{1}{\sigma }f_{0}\left(\frac{x-\mu }{\sigma }\right),$$
where *μ* is the location parameter, *σ* > 0 is the scale parameter, and *f*_0_ is the standard density of the distribution *Z* ∼ *F*(*x*; 0, 1). This implies that for any *X* ∼ *F*(*x*; *μ*, *σ*), the standardized variable $Z=\frac{\left(X-\mu \right)}{\sigma }$ follows the standard distribution *F*(*x*; 0, 1).

A key property of this family is its invariance under affine transformations, implying that if *X* ∼ *F*(*x*; *μ*, *σ*), then *a* + *bX* ∼ *F*(*x*; *a* + *bμ*, |*b*|*σ*). The normal distribution is a classic example of the location-scale distribution. Other prominent examples include the Cauchy and logistic distributions.
Cauchy distribution: Its pdf is given by$$\begin{array}{c} f(x;\theta ,\lambda )=\frac{1}{\pi \lambda \left[1+{\left(\frac{x-\theta }{\lambda }\right)}^{2}\right]}, \end{array}$$
where *θ* is the location parameter and *λ* is the scale parameter. Note that the Cauchy distribution does not have a defined mean or variance.Logistic distribution: Named for its relationship to the logistic (sigmoid) function, the logistic distribution resembles the normal distribution in shape but possesses heavier tails. It finds applications in logistic regression, growth models, and economics. Its cumulative distribution function (cdf) is given by$$\begin{array}{c} F(x;\alpha ,\beta )=\frac{1}{1+\exp(-(x-\alpha )/\beta )},\;x,\;\alpha \in R,\;\beta >0. \end{array}$$
There are more other distributions in this category. For further reading, see [9,10].

#### 2.1.2. Pearson Distribution Family

The Pearson family is a system of continuous probability distributions, introduced by Karl Pearson [11], designed to model a wide range of empirical data characterized by properties such as skewness and kurtosis that deviate from the normal distribution. The system is defined by the Pearson differential equation:(2)$$\begin{array}{c} \frac{df(x)}{dx}=f\left(x\right)\frac{x-\lambda }{ax^{2}+bx+c}, \end{array}$$
where *f*(*x*) is a probability density function and the parameters *a*, *b*, *c*, *λ* determine the specific distribution type.

Roots of the denominator (*ax*^2^ + *bx* + *c*) classify distributions into eight or more types listed in Table 1, where type 0 is the normal distribution; type I distribution is the beta distribution; type II distribution curve is a symmetrical *U*-shaped curve; type III distribution curve is the gamma distribution; type IV distribution curve is skewed heavy-tailed; type V is the shift inverse gamma distribution; type VI is an inverse beta distribution; type VII is the *t*-distribution. Some references consider more types, like types VIII and IX, which are special cases of the power function distribution, and type X, which is the exponential distribution. More details can be found in Chapter 12 of [1,12].

#### 2.1.3. The Exponential Distribution Family

Consider it as a “family” in the biological sense, implying that though its members, such as the normal, exponential, gamma, beta, and Poisson distributions, may appear distinct characteristics, they share a common underlying mathematical structure. This shared “DNA” makes them exceptionally powerful and convenient to use in statistical modeling and machine learning.

Formally, the exponential family comprises distributions whose probability density (or mass) functions can be expressed in the following canonical form: (3)$$\begin{array}{c} f\left(x;\theta \right)=h\left(x\right)\exp\left\{\theta^{⊤}T\left(x\right)-B\left(\theta \right)\right\}, \end{array}$$
where *h*(*x*) is a function as base measure that depends only on the data *X*, not the parameters $\theta ={\left(\theta_{1},\ldots ,\theta_{k}\right)}^{⊤}$. $T\left(x\right)={\left(T_{1}\left(x\right),\ldots ,T_{k}\left(x\right)\right)}^{⊤}$ is a sufficient statistic, and *B*(***θ***) is a log-partition function and a normalization constant that ensures that the function integrates/sums to 1. *B*(***θ***) also generates the moments (mean, variance) of the distribution.

This common structure leads to several powerful properties:(1)All members have a low-dimensional sufficient statistic **T**(*x*).(2)In Bayesian statistics, the exponential family always has a natural conjugate prior. This makes Bayesian updating mathematically neat and computationally tractable.(3)Distributions in the exponential family are the maximum entropy distributions given constraints on the expected values of the sufficient statistics. For example, the normal distribution has the maximum entropy for a given mean and variance.

Further details can be found in references such as [1,9]. Many additional families of univariate distributions have also been cataloged in [1].

### 2.2. Distributions Produced by a Function of Random Variables

When the distribution of a random variable *X* or a random vector $X=(X_{1},\ldots ,X_{k})$ is known, applying a continuous function *Y* = *g*(*X*) or *Y* = *h*(***X***) in order to yield a new distribution is a prevalent distribution construction approach. In this context, the original distribution is termed the initial distribution, while the resulting distribution is called the target distribution. The inverse transformation method [13] is a widely used technique for statistical simulation, based on the following theorem.

**Theorem** **1.***Let X have a continuous cdf F*(*x*) *and a uniform random variable U ∼ U*(*0, 1*)*. Then we have**(a)* *F*^−1^(*U*) ∼ *F*(·);*(b)* *F*(*X*) ∼ *U*(0, 1).

Following Theorem 1, we can leverage this connection between any continuous distribution and the uniform distribution to derive the distribution of a transformed random variable *Y* = *g*(*X*). The key insight is to use the probability integral transform from part (b) of the theorem. We know that *F*_*X*_(*X*) ∼ *U*(0, 1). Therefore, the cdf of the transformation *Y* = *g*(*X*) can be found by relating the event {*Y* ≤ *y*} back to this uniform distribution. Specifically, the cdf of *Y* is$$F_{Y}\left(y\right)=P\left(Y\leq y\right)=P\left(g\left(X\right)\leq y\right).$$
The solution to this probability depends on the nature of the function *g*. For monotonic and differentiable transformations, this approach leads directly to the well-known change of variables formula for the pdf of *Y*:$$f_{Y}\left(y\right)=f_{X}\left(x\right)\left|\frac{dx}{dy}\right|=f_{X}\left(g^{-1}\left(y\right)\right)\left|\frac{d}{dy}g^{-1}\left(y\right)\right|,$$
where *x* = *g*^−1^(*y*) is the inverse function. This formula can be derived by differentiating the cdf *F*_*Y*_(*y*), which itself is computed using the distribution of *X* that Theorem 1 helps us to understand and manipulate through the uniform distribution.

A systematic study and generalization of the inverse transformation method can be found in Li et al. [14]. Numerous distributions have been constructed using this approach. The following are some functions commonly employed in the literature for this purpose.

#### 2.2.1. Transformation of Random Variables

Some transformation examples are presented later.

Linear transformation *Y* = *a* + *bX*, for example, used in the location-scale distributions;Power transformation *Y* = *X*^*r*^, for example, from the normal to the *χ*^2^ distribution by *Y* = *X*^2^;Exponential transformation *Y* = *e*^*X*^, for example, from the normal to the lognormal distribution by *Y* = *e*^*X*^;Inverse transformation *Y* = 1/*X*, for example, the inverse gamma distribution.Transformation $Y=g(X_{1},\ldots ,X_{m})$, for example, the *F*-distribution and beta distribution are produced by functions of two *χ*^2^ distribution variables.

#### 2.2.2. Order Statistics

These include fundamental objects in statistics, for example, the sample minimum, maximum, and median. They deal with the properties and applications of sorted random samples. Let $X_{1},\ldots ,X_{n}$ be a random sample from a population with cdf *F* and pdf *f*. The order statistics, denoted by $X_{\left(1\right)}\leq X_{\left(2\right)}\leq \cdots \leq X_{\left(n\right)}$, represent the sorted values of the sample. The cdf of the *k*th order statistic, *X*_(*k*)_, is given by(4)P(X(k)≤x)=∑j=knnj[F(x)]j[1−F(x)]n−j.
The density functions for specific order statistics are provided in Table 2. Order statistics have wide-ranging applications, including robust statistics, sample range and interquartile range, quantiles and Q–Q plots, reliability engineering, and non-parametric statistical tests (e.g., the Wilcoxon rank-sum test). A comprehensive introduction to order statistics can be found in [15].

#### 2.2.3. Extreme Value Theory (EVT)

EVT is a statistical framework for assessing the probabilities of extreme events. The fundamental result in EVT is the extreme value theorem, stating that the distribution of properly normalized maxima (or minima) converges to one of three types of distributions, regardless of the underlying distribution of the original data. These three types are collectively known as the generalized extreme value (GEV) distribution. The cdf for the GEV is given by(5)$$\begin{array}{c} F(x)=\exp\left\{-{\left[1+\xi \left(\frac{x-\mu }{\sigma }\right)\right]}^{-1/\xi }\right\},\mathrm{if}1+\xi \frac{x-\mu }{\sigma }>0, \end{array}$$
where parameters are *μ* for location, *σ* > 0 for scale, and *ξ* for shape. The three special cases of the GEV distribution are

Type I (Gumbel distribution) *Gu*(*μ*, *σ*) light-tailed, with *ξ* = 0. Its pdf is$$\frac{1}{\sigma }\exp\left\{-\frac{x-\mu }{\sigma }-\exp\left\{-\frac{x-\mu }{\sigma }\right\}\right\},\sigma >0.$$Type II (Fr$\overset{´}{e}$chet distribution) heavy-tailed, with *ξ* > 0.Type III (Weibull distribution) short-tailed and bounded with *ξ* < 0.

The extreme value distribution has been used in finance, environmental science, engineering, and reliability testing. For a rigorous treatment of the GEV distribution and extreme value theory, see [16].

### 2.3. Generalized Families via Transformation of the cdf

Given a baseline cdf *F*(*x*), the exponentiated (or power-transformed) family is defined by$$G\left(x\right)={\left[F\left(x\right)\right]}^{\alpha },\;\alpha >0.$$
This construction, detailed in [17], generates a vast range of new distributions. If $X_{1},X_{2},\ldots ,X_{n}$ is a random sample from *F*(*x*), then *G*(*x*) is the cdf of the sample maximum, $\max(X_{1},\ldots ,X_{n})$, when *α* = *n* is an integer. Notable members include the exponentiated Weibull and exponentiated exponential distributions.

A dual transformation yields the following complementary exponentiated family:$$G\left(x\right)=1-{\left[1-F\left(x\right)\right]}^{\beta },\;\beta >0.$$
This is the cdf of the sample minimum, $\min(X_{1},\ldots ,X_{n})$, when *β* = *n*. It is also known as the proportional odds model in survival analysis. Together, the exponentiated and complementary exponentiated families form the foundation of the broader class of maximum–minimum stable distributions.

Further generalizations have been proposed to introduce additional flexibility and skewness. The Alpha Power Family (or T-X family), defined by$$G\left(x\right)=\frac{\alpha^{F(x)}-1}{\alpha -1},\;\alpha >0,\alpha \neq 1,$$
effectively introduces skewness to the baseline distribution [18]. A highly flexible generalization is the Kumaraswamy-G family [19], given by$$G\left(x\right)=1-{\left[1-F{\left(x\right)}^{a}\right]}^{b},\;a>0,b>0.$$
This family, which contains the exponentiated family as a special case (*b* = 1), enjoys the advantage of having a quantile function in a simple closed form.

These methods, along with other systematic frameworks like the T-X family [20], demonstrate that simple transformations of a baseline cdf can serve as an engine for generating numerous new and useful distributions, greatly expanding the toolbox for statistical modeling.

### 2.4. Combination Methods

There are several approaches to constructing a distribution through combining the existing distributions.

#### 2.4.1. Types of Mixtures

Finite mixture $f\left(x\right)=\sum_{i=1}^{k}w_{i}f_{i}\left(x\right)$, which has various applications in statistical inference with flexibility in density fitting [21], for example, a mixture of two normals [22] or two Weibull distributions [23].Continuous mixture *f*(*x*) = ∫ *f*(*x*; *θ*)*g*(*θ*)*dθ*. One example is the scale mixture. A primary method to create such continuous mixtures is through compounding, which we detail in the next section.

The mixing distribution *g*(*θ*) in an infinite mixture is not required to be continuous. When *g*(*θ*) is a discrete distribution, the resulting mixture, often called a countable mixture or discrete mixture, integrates over a countable set of parameters. This approach is technically distinct from the continuous mixture defined above but remains a powerful tool for deriving flexible distributions. Prominent examples of distributions derived from a discrete mixing mechanism include the following:The negative binomial distribution, which can be derived as a Poisson-gamma mixture where the Poisson rate parameter is governed by a gamma distribution. While the gamma is continuous, a discrete analogue exists (e.g., a Poisson mixture with a discrete log-gamma distribution would yield a similar over-dispersed count distribution).The Student’s *t*-distribution, which is a scale mixture of normal distributions where the precision (inverse variance) parameter follows a gamma distribution.The beta-binomial distribution, which arises when the success probability *p* of a binomial distribution follows a beta distribution.The compound Poisson distribution and related models, where the intensity or other parameters are driven by a latent discrete process.

#### 2.4.2. Compounding

It is a fundamental concept in probability theory that a parameter of one distribution is itself a random variable following another distribution (see [1]). This approach generates a more flexible distribution that incorporates uncertainty in the parameter. It is important to note that the marginal distribution resulting from compounding is mathematically equivalent to a continuous mixture distribution, as defined in Section 2.4.1. The compounding framework provides a hierarchical data-generation perspective for the same model. Formally, if *X*|*θ* ∼ *F*_1_(*θ*) and *θ* ∼ *F*_2_(*λ*), then the distribution of *X* is the marginal distribution of a compound distribution. Examples are the beta-binomial, gamma-Poisson (negative binomial), normal–normal, Dirichlet–multinomial, compound Poisson process models. The normal–normal model is as follows:First stage (conditional distribution): $X|\mu ∼N(\mu ,\sigma^{2})$. The observation *X* is normally distributed with mean *μ* and a known variance *σ*^2^;Second stage (parameter distribution): $\mu ∼N(\mu_{0},\tau^{2})$. The mean *μ* itself is assumed to be normally distributed with prior mean *μ*_0_ and variance *τ*^2^;Compound distribution: The unconditional distribution of $X∼N(\mu_{0},\sigma^{2}+\tau^{2})$.

#### 2.4.3. Convolution

The random variable *Z* = *X* + *Y* is termed the convolution of *X* and *Y*, and it has a density function given by$$f_{X+Y}\left(z\right)=\int_{-\infty }^{+\infty }f_{X,Y}\left(x,z-x\right)dx,$$
where *f*_*X*,*Y*_(·, ·) is the joint density of (*X*, *Y*). If *X* and *Y* are independent, this simplifies to $f_{X+Y}\left(z\right)=\left(f_{X}*f_{Y}\right)\left(z\right)=\int_{-\infty }^{+\infty }f_{X}\left(x\right)f_{Y}\left(z-x\right)dx$.

A distribution of significant practical importance arises when *X* follows a normal distribution *N*(*μ*, *σ*^2^) and *Y* follows an exponential distribution Exp(*λ*), with *X* and *Y* being independent. This specific convolution forms the cornerstone of the normal–exponential stochastic frontier model [24], widely used in econometrics to measure the efficiency of firms and other decision-making units. In this context, the normal random variable represents the random noise (e.g., measurement error, luck), while the exponential random variable captures one-sided inefficiency.

Then the density function of *Z* = *X* + *Y* is given by$$f_{Z}\left(z;\mu ,\sigma ,\lambda \right)=\frac{\lambda }{2}e^{\frac{\lambda }{2}\left(2\mu +\lambda \sigma^{2}-2z\right)}\cdot \mathrm{erfc}\left(\frac{\mu +\lambda \sigma^{2}-z}{\sqrt{2}\sigma }\right),$$
where erfc(*z*) represents the complementary error function defined as$$\mathrm{erfc}\left(z\right)=\frac{2}{\sqrt{\pi }}\int_{z}^{\infty }e^{-t^{2}}dt.$$

### 2.5. Some Key Univariate Distributions and Related Distributions

The first continuous statistical distribution to be formally studied was the uniform distribution, dating back to Huygens [25]. However, the most historically significant early continuous distribution is the normal distribution, also referred to as the Gaussian distribution. Although the uniform distribution provided the simplest continuous model, it was the normal distribution that first underwent extensive analytical study. The development of the normal distribution emerged later through the work of de Moivre [26] and Gauss [27] during the 18th and 19th centuries. Its fundamental role in error theory and the central limit theorem established it as the cornerstone of continuous probability theory.

#### 2.5.1. Distributions Related to the Normal Distribution

The probability density function of a normal distribution *X* ∼ *N*(*μ*, *σ*^2^) is given by(6)$$\begin{array}{c} f\left(x;\mu ,\sigma \right)=\frac{1}{\sqrt{2\pi }\sigma }e^{-\frac{1}{2}{\left(\frac{x-\mu }{\sigma }\right)}^{2}},-\infty <x<\infty , \end{array}$$
where parameters *μ* and *σ*^2^ are the mean and variance of *X*, respectively. When *μ* = 0 and *σ*^2^ = 1, the distribution is referred to as the standard normal distribution, denoted by *Z* ∼ *N*(0, 1). Several distributions derived from the normal distribution are listed in Table 3 and can be constructed as follows:(1)Lognormal distribution: *X* ∼ *LN*(*μ*, *σ*^2^) if ln(*X*) ∼ *N*(*μ*, *σ*^2^);(2)Skew-normal distribution: proposed by Azzalini [28] with the pdf(7)$$\begin{array}{c} f(z;\alpha )=2\phi (z)\Phi (\alpha z),z\in R,\alpha \in R, \end{array}$$
where *ϕ*(*z*) is the standard normal pdf, Φ(*αz*) is is the standard normal cdf evaluated at *αz* and *α* is a shape parameter. For more general cases, refer to Table 3. Skew-normal distribution is one of the skew-elliptical distributions, and there are more such distributions including skew-*t*, skew-Cauchy, skew-logistic, and skew-Laplace distributions; see [29,30]. There is more discussion about this topic in Section 4.6.(3)Chi-square distribution: If $X_{1},\ldots ,X_{\nu }$ are independent and identically distributed (*i*.*i*.*d*.) random variables of *N*(0, 1), then $X=X_{1}^{2}+\ldots +X_{\nu }^{2}∼\chi^{2}\left(\nu \right)$, the *χ*^2^-distribution with degrees of freedom *ν*;(4)Chi distribution: If $X_{1},\ldots ,X_{\nu }$ are *i*.*i*.*d*. *N*(0, 1), then $X=\sqrt{X_{1}^{2}+\ldots +X_{\nu }^{2}}∼\chi \left(\nu \right)$ the *χ*-distribution with degrees of freedom *ν*;(5)Beta distribution (special case): If *Y* ∼ *χ*^2^(2*m*) and *Z* ∼ *χ*^2^(2*n*) are independent, then $X=\frac{Y}{Y+Z}∼Be\left(m,n\right)$, referred to Table 3;(6)Beta prime distribution: If *X* ∼ *Be*(*α*, *β*), then $\frac{X}{1-X}∼BP\left(\alpha ,\beta \right)$;(7)Arcsine distribution: It is defined as $Be(\frac{1}{2},\frac{1}{2})$;(8)*F*-distribution: If *Y* ∼ *χ*^2^(*m*) and *Z* ∼ *χ*^2^(*n*) are independent, then $X=\frac{nY}{mZ}∼F\left(m,n\right)$;(9)*t*-distribution: If *Y* ∼ *χ*^2^(*ν*) and *Z* ∼ *N*(0, 1) are independent, then $T=\sqrt{\nu }\frac{Z}{\sqrt{Y}}∼t_{\nu }$.

**Table 3 entropy-27-01188-t003:** Some continuous distributions constructed by or closely related to *N*(*μ*, *σ*^2^).

Distribution	Notation	Density f(x)
Normal	*N*(*μ*, *σ*^2^)	$\frac{1}{\sqrt{2\pi }\sigma }\exp\left\{-\frac{{\left(x-\mu \right)}^{2}}{2\sigma^{2}}\right\},x,\mu \in R,\sigma >0$
Lognormal	*LN*(*μ*, *σ*^2^)	$\frac{1}{x\sigma \sqrt{2\pi }}\exp\left(-\frac{{\left(\ln x-\mu \right)}^{2}}{2\sigma^{2}}\right)$, x>0,μ∈R,σ>0
Skew-normal	*SN*(*μ*, *σ*, *α*)	$f\left(x;\mu ,\sigma ,\alpha \right)=\frac{2}{\sigma }\phi \left(\frac{x-\mu }{\sigma }\right)\Phi \left(\alpha \left(\frac{x-\mu }{\sigma }\right)\right),\sigma >0,x,\mu ,\alpha \in R$
*χ*^2^(*ν*)	*χ*^2^(*ν*)	${\left[2^{\nu /2}\Gamma \left(\frac{\nu }{2}\right)\right]}^{-1}x^{\nu /2-1}e^{-x/2},x>0,\nu >0$
*χ*(*ν*)	*χ* _ *ν* _	${\left[2^{\nu /2-1}\Gamma \left(\frac{\nu }{2}\right)\right]}^{-1}x^{\nu -1}e^{-x^{2}/2},x>0,\nu >0$
Beta	*Be*(*a*, *b*)	$\frac{1}{B(a,b)}x^{a-1}{\left(1-x\right)}^{b-1},0\leq x\leq 1,a,b>0$
Beta prime	*BP*(*α*, *β*)	$\frac{1}{B(\alpha ,\beta )}\frac{x^{\alpha -1}}{{\left(1+x\right)}^{\alpha +\beta }},x>0,\alpha ,\beta >0$
Arcsine	arcsine	$\frac{1}{\pi \sqrt{x(1-x)}},0<x<1$
*F*	*F*(*m*, *n*)	$\frac{1}{B(\frac{m}{2},\frac{n}{2})}{\left(\frac{m}{n}\right)}^{\frac{m}{2}}x^{\frac{m}{2}-1}{\left(1+\frac{m}{n}x\right)}^{-\frac{m+n}{2}},x>0,m,n>0$
Student-*t*	*t* _ *ν* _	$\frac{\Gamma (\frac{\nu +1}{2})}{\sqrt{\nu \pi }\Gamma \left(\frac{\nu }{2}\right)}{\left(1+\frac{x^{2}}{\nu }\right)}^{-\frac{\nu +1}{2}},\nu >0,x\in R$

Note: The title “constructed by” implies a direct derivation from the normal distribution. For distributions like *χ*^2^, *χ*, *F*, and *t*, this direct construction (e.g., as a sum of squares of independent standard normal variables) is most straightforward when their shape/degrees of freedom parameters (*v*, *m*, *n*) are positive integers. The density functions listed here represent the analytic continuations of these distributions to positive real-valued parameters, which are their standard definitions. The beta and arcsine distributions are included due to their relationship with normal random variables via the transformation of ratios or functions thereof.

#### 2.5.2. Distributions Related to the Exponential Distribution

The exponential distribution, defined on the positive real number space, was extensively studied in the early 19th century due to its valuable properties, such as memorylessness and its connection to the Poisson process for modeling rare events (see [1]). Several distributions derived from the exponential distribution are listed in Table 4.
(1)Laplace distribution: Also known as the double exponential distribution, the Laplace distribution extends the exponential distribution symmetrically to the entire real number space, R.(2)Weibull distribution: The Weibull distribution is formed by introducing location *δ* and sharp *α* parameters to the density of the exponential density. It constitutes a flexible family of distributions widely used for modeling time-to-failure, lifespan, and other positive-valued measurements.(3)Rayleigh distribution: Proposed by Rayleigh [31] for modeling wave amplitudes, the Rayleigh distribution is a special case of the Weibull distribution with *Wei*(2, *β*, 0). It also arises from the normal distribution, *N*(0, *σ*^2^), by $X=\sqrt{Y^{2}+Z^{2}}∼Ray\left(\sigma \right)$, where *Y* and *Z* are *i*.*i*.*d*. *N*(0, *σ*^2^) and *β* = 2*σ*^2^.(4)Topp–Leone exponential distribution (TLED): Introduced by [32], the TLED adds a shape parameter *α* to the exponential distribution, yielding the cumulative distribution function $F\left(x\right)={\left(1-e^{-2x/\beta }\right)}^{\alpha }$. Its density is provided in Table 4. The TLED generalizes certain distributions. For instance, TLED includes the Burr type as *TLED*(2, *β*) and Frèchet distribution as *TLED*(1/2, *β*).

**Table 4 entropy-27-01188-t004:** Some continuous distributions constructed by the exponential distribution.

Distribution	Notation	Density p(x)
Exponential	*Exp*(*β*)	$\frac{1}{\beta }e^{-x/\beta },\;x\geq 0,\beta >0$
Laplace	*Laplace*(*μ*, *β*)	$\frac{1}{2\beta }e^{-\frac{\left|x-\mu \right|}{\beta }},\;x\in R,\;\mu \in R,\;\beta >0$
Weibull	*Wei*(*α*, *β*, *δ*)	$\frac{\alpha }{\beta }{\left(\frac{x-\delta }{\beta }\right)}^{\alpha -1}e^{-{\left(\frac{x-\delta }{\beta }\right)}^{\alpha }},\;x\geq \delta ,\;\delta \in R,\;\alpha ,\beta >0$
Rayleigh	*Ray*(*β*) or *Ray*(*σ*)	$\frac{2x}{\beta }e^{-x^{2}/\beta },\;x\geq 0,\;\beta =2\sigma^{2}$
TLED	*TLED*(*α*, *β*)	$\frac{2\alpha }{\beta }e^{-2x/\beta }{\left(1-e^{-2x/\beta }\right)}^{\alpha -1},\;x>0,\;\alpha ,\beta >0$

It is worth mentioning that the exponential distribution and the Poisson distribution share a fundamental symbiotic relationship within the context of a Poisson process. If the number of events occurring in a fixed time interval follows a Poisson distribution with rate parameter *λ*, then the waiting time between consecutive events follows an exponential distribution with the same rate parameter $\lambda =\frac{1}{\beta }$. Thus, the former models event counts, and the latter models inter-arrival times.

#### 2.5.3. Distributions Related to the Gamma Distribution

The gamma distribution, *Gamma*(*α*, *β*), was formally defined in the 20th century and can be considered as an extension of the exponential distribution. It retains several desirable properties of the exponential distribution, such as additivity: the sum of independently and identically distributed variables *X*_*i*_ ∼ *Gamma*(*α*_*i*_, *β*) for i=1,…,m is distributed as $Gamma(\sum_{i=1}^{m}\alpha_{i},\beta )$. The gamma distribution is foundational, as the chi-square distribution $\chi_{\nu }^{2}$ is a special case, *Gamma*(*ν*/2, 1/2), and the beta distribution *Be*(*a*, *b*) can be defined by $\frac{Y}{Y+Z}$, where *Y* ∼ *Gamma*(*a*, 1) and *Z* ∼ *Gamma*(*b*, 1) are independent. Table 5 lists several important extensions of the gamma distribution.
(1)Location-shift gamma: Adding a location parameter *δ* forms the three-parameter distribution *Gamma*(*α*, *β*, *δ*).(2)Generalized gamma: This distribution exists in several forms. The version shown with shape *a*, scale *d*, and power *p* parameters was introduced by Stacy [33]. It includes several common distributions as special cases: the standard gamma (*p* = 1), the Weibull (*a* = 1), the Rayleigh (*a* = *p* = 2), and the exponential (*a* = *p* = 1).(3)Inverse gamma: Defined as *Y* = 1/*X* for *X* ∼ *Gamma*(*α*, *β*), this two-parameter distribution is positive-skewed. It is primarily used in Bayesian statistics as the conjugate prior for the variance of a normal distribution.(4)Beta prime distribution: The ratio *X*/*Y* of two independent variables, *X* ∼ *Gamma*(*α*, *θ*) and *Y* ∼ *Gamma*(*β*, *θ*), follows a beta prime distribution *BP*(*α*, *β*).(5)General beta prime distribution: This is a further extension of the beta prime distribution, incorporating two additional parameters.

**Table 5 entropy-27-01188-t005:** Some continuous distributions constructed by the gamma distribution.

Distribution	Notation	Density f(x)
Gamma	*Gamma*(*α*, *β*)	$\frac{\beta^{\alpha }}{\Gamma (\alpha )}x^{\alpha -1}e^{-\beta x},\;x>0,\;\alpha ,\beta >0$
Location-shift gamma	*Gamma*(*α*, *β*, *δ*)	$\frac{\beta^{\alpha }}{\Gamma (\alpha )}{\left(x-\delta \right)}^{\alpha -1}e^{-\beta (x-\delta )}$, x≥δ,δ∈R, *α*, *β* > 0
Generalized gamma	*G* − *gamma*(*a*, *d*, *p*)	$\frac{p}{d^{a}\Gamma \left(a/p\right)}x^{a-1}e^{-{\left(x/d\right)}^{p}},\;x>0,\;a,d,p>0$
Inverse gamma	*Inv* − *Gamma*(*α*, *β*)	$\frac{\beta^{\alpha }}{\Gamma (\alpha )}x^{-(\alpha +1)}e^{-\beta /x},\;x>0,\;\alpha ,\beta >0$
Beta prime	*BP*(*α*, *β*)	$\frac{1}{B(\alpha ,\beta )}\frac{x^{\alpha -1}}{{\left(1+x\right)}^{\alpha +\beta }},\;x>0,\;\alpha ,\beta >0$
General beta prime	*GBP*(*α*, *β*, *p*, *q*)	$\frac{\left|\alpha \right|{\left(1+{\left(x/\beta \right)}^{\alpha }\right)}^{-p-q}{\left(x/\beta \right)}^{p\alpha -1}}{\beta B(p,q)}$, α∈R,α≠0, *p*, *q*, *β* > 0, *x* > 0.

### 2.6. The Generalized Hyperbolic Distribution Family

Schrdinger [34] described the probability distribution of the first passage time in Brownian motion. Thirty years later, Tweedie [35] gave the inverse relationship between the cumulant generating function of the first passage time distribution and that of the normal distribution, and named it the inverse Gaussian (IG) distribution. This new distribution was proposed in the top journal Nature, which shows its importance. The IG distribution, denoted as *IG*(*μ*, *λ*), is defined for *x* > 0, *μ* > 0, and *λ* > 0 by the probability density function:(8)$$\begin{array}{c} f\left(x;\mu ,\lambda \right)=\sqrt{\frac{\lambda }{2\pi x^{3}}}\exp\left\{\frac{-\lambda {\left(x-\mu \right)}^{2}}{2\mu^{2}x}\right\}. \end{array}$$
The IG distribution models the first passage time of a Brownian motion, making it highly useful for analyzing lifetime data and the frequency of event occurrences in various fields. For a comprehensive reference, see [36]. The IG distribution has many extensions, among which the generalized hyperbolic distribution (GH) is a more general one.

The GH distribution is a continuous probability distribution that represents a highly flexible family of models for describing data with skewness and heavy tails. Its key strength lies in its ability to nest several other important distributions as special or limiting cases, providing a unified framework for statistical modeling. It was introduced by [37] to model the grain size distributions of wind-blown sand. However, its remarkable fit to financial data catapulted it to prominence in financial econometrics and risk management (see [38]).

A random variable *X* is said to follow a GH distribution *GH*(*λ*, *α*, *β*, *δ*, *μ*) if its pdf is(9)$$\begin{array}{c} f\left(x\right)=C\left(\lambda ,\alpha ,\beta ,\delta \right){\left(\delta^{2}+{\left(x-\mu \right)}^{2}\right)}^{(\lambda -0.5)/2}K_{\lambda -0.5}\left(\alpha \sqrt{\delta^{2}+{\left(x-\mu \right)}^{2}}\right)\exp\left(\beta \left(x-\mu \right)\right), \end{array}$$
where x∈R and$$\begin{array}{c} C\left(\lambda ,\alpha ,\beta ,\delta \right)=\frac{{\left(\alpha^{2}-\beta^{2}\right)}^{\lambda /2}}{\sqrt{2\pi }\alpha^{\lambda -0.5}\delta^{\lambda }K_{\lambda }\left(\delta \sqrt{\alpha^{2}-\beta^{2}}\right)},\;\alpha >0,\;\left|\beta \right|<\alpha ,\;\delta >0,\;\mu ,\lambda ,\in R. \end{array}$$
The parameter *λ* influences the tail behavior and the subfamily classification; *α* controls the heaviness of the tails and the steepness; *β* controls the asymmetry; *δ* is the scale parameter related to the dispersion; and *μ* is the location parameter. *K*_*λ*_(·) is the modified Bessel function of the second kind of order *λ* defined by$$\begin{array}{c} K_{\lambda }\left(x\right)=\frac{1}{2}\int_{0}^{\infty }t^{\lambda -1}\exp\left(-\frac{x}{2}\left(t+\frac{1}{t}\right)\right)dt. \end{array}$$
The GH distribution involves many distributions as its special cases: hyperbolic distribution (*λ* = 1), normal inverse Gaussian distribution (*λ* = −0.5), variance gamma distribution (*λ* > 0, *δ* = 0), etc. A normal variance-mean mixture distribution, termed the normal inverse Gaussian distribution proposed by [39], is used to construct stochastic processes that appear of interest for statistical modeling purposes, particularly in turbulence and finance. Its pdf is defined as$$\begin{array}{c} f\left(x;\alpha ,\beta ,\delta ,\mu \right)=C\left(\alpha ,\beta ,\delta \right)q{\left(\frac{x-\mu }{\delta }\right)}^{-1}K_{1}\left(\alpha \delta q\left(\frac{x-\mu }{\delta }\right)\right)\exp\left(\beta \left(x-\mu \right)\right),\;x\in R, \end{array}$$
where$$\begin{array}{c} C\left(\alpha ,\beta ,\delta \right)=\frac{\alpha }{\pi }\exp\left(\delta \sqrt{\alpha^{2}-\beta^{2}}\right),\;q\left(t\right)=\sqrt{1+t^{2}}, \end{array}$$
and *K*_1_ is the modified Bessel function of the second kind of order 1. The parameters have the following role: μ∈R (location) and *δ* > 0 (scale); *α* > 0 (tail heaviness) controls the steepness of the distribution. A larger *α* leads to lighter tails and *α* > |*β*|, where *β* (asymmetry/skewness) controls the skewness. *β* = 0 gives a symmetric distribution, *β* > 0 implies positive skewness, and *β* < 0 implies negative skewness.

The GH distribution is a normal variance-mean mixture of the generalized inverse Gaussian (GIG) distributions. This construction is the source of its flexibility and mathematical tractability. The GIG distribution is a continuous probability distribution that generalizes several important distributions and plays a critical role as a mixing distribution in normal variance-mean mixtures. The GIG distribution denoted by *GIG*(*λ*, *χ*, *ψ*) is defined by the given pdf(10)$$\begin{array}{c} f\left(x;\lambda ,\chi ,\psi \right)=\frac{{\left(\psi /\chi \right)}^{\lambda /2}}{2K_{\lambda }\left(\sqrt{\psi \chi }\right)}x^{\lambda -1}\exp\left(-\frac{1}{2}\left(\frac{\chi }{x}+\psi x\right)\right),\;x>0,\;\chi ,\psi \geq 0, \end{array}$$
where parameter *λ* determines the shape of the distribution and is deeply connected to the index of the Bessel function. *χ* is the scale parameter often related to the “left-hand” behavior, with the constraint: *λ* > 0 whenever *χ* = 0. *ψ* is another scale parameter that is often related to the “right-hand” tail with the constraint: *λ* < 0 whenever *ψ* = 0, and *ψ* = *χ* = 0 is not allowed. The GIG involves inverse Gaussian distribution (*λ* = −0.5), reciprocal inverse Gaussian distribution (*λ* = 0.5), and gamma distribution (*χ* = 0, *λ* > 0).

A random variable *Y* follows a GH distribution *GH*(*λ*, *α*, *β*, *δ*, *μ*) if it has the following stochastic representation:$$\begin{array}{c} Y=\mu +\beta X+\sigma \sqrt{X}Z, \end{array}$$
where *Z* ∼ *N*(0, 1), and *X* ∼ *GIG*(*λ*, *χ*, *ψ*) is independent of *Z*. The parameters of the resulting *GH*(*λ*, *α*, *β*, *δ*, *μ*) distribution are identified as follows: $\alpha =\sqrt{\psi \sigma^{2}+\beta^{2}},\delta =\sqrt{\chi }$. By choosing different GIG subfamilies as the mixer, we obtain the different GH subfamilies:(1)Mixing with the inverse Gaussian (*λ* = −0.5) distribution results in the normal inverse Gaussian distribution.(2)Mixing with the gamma (*χ* = 0, *λ* > 0) distribution results in the variance gamma distribution.(3)Mixing with the GIG (*λ* = 1) distribution results in the hyperbolic distribution.

### 2.7. Beta-Generated Distributions

A significant research area is the creation of flexible univariate distribution classes that extend classical models. The beta-generated (BG) distributions by Arnold et al. [40,41] achieve this by merging the beta distribution with a chosen baseline distribution.

**Definition** **1.***For a baseline cdf F*(*x*) *with pdf f*(*x*) *on a domain D, the beta-generated (BG) distributions have the following pdf:*(11)$$\begin{array}{c} g_{F}\left(x;a,b\right)=\frac{1}{B(a,b)}F{\left(x\right)}^{a-1}{\left[1-F\left(x\right)\right]}^{b-1}f\left(x\right),\;x\in D,\;a>0,b>0. \end{array}$$*We denote this by X ∼ BG*(*a*, *b*; *F*)*. The corresponding cdf is derived via the substitution y *= *F*(*t*)*:*$$\begin{array}{c} G_{F}\left(x;a,b\right)=\frac{1}{B(a,b)}\int_{0}^{F(x)}y^{a-1}{\left(1-y\right)}^{b-1}dy=I_{F(x)}\left(a,b\right), \end{array}$$*which is the incomplete beta function ratio.*

A key property of the *BG*(*a*, *b*; *F*) distribution is its stochastic representation:(12)$$\begin{array}{c} X\overset{d}{=}F^{-1}\left(B\right),\;\mathrm{for}\;B∼Be\left(a,b\right). \end{array}$$
This representation provides a foundation for applying the generalized inverse transformation method in statistical simulation using representative points. It offers a direct method for simulating from the density (Equation 11) and for constructing multivariate extensions. The moments of *X* are consequently given by(13)$$\begin{array}{c} E\left(X^{r}\right)=E({\left[F^{-1}\left(B\right)\right]}^{r}),\;r>0. \end{array}$$
The beta-generated family encompasses a wide range of distributions depending on the choice of the baseline *F*. To demonstrate the flexibility and utility of this class, we select two specific examples. For the generalization of this generating mechanism, see [42].

#### 2.7.1. The Beta-Normal Distribution

If *F*(*x*) is the cdf of *N*(*μ*, *σ*^2^), the corresponding BG distribution is the beta-normal distribution, denoted as *X* ∼ *BN*(*a*, *b*; *μ*, *σ*). As studied by Eugene et al. [43], its cdf is(14)$$\begin{array}{c} G\left(x\right)=\frac{1}{B(a,b)}\int_{0}^{\Phi (\frac{x-\mu }{\sigma })}t^{a-1}{\left(1-t\right)}^{b-1}dt, \end{array}$$
where Φ(·) is the standard normal cdf. For the standard case *BN*(*a*, *b*; 0, 1), the pdf is(15)$$\begin{array}{c} g\left(x\right)=\frac{1}{B(a,b)}\Phi {\left(x\right)}^{a-1}{\left[1-\Phi \left(x\right)\right]}^{b-1}\phi \left(x\right),\;x\in R. \end{array}$$
Its moments are given by the formula in (Equation 13) (Gupta and Nadarajah [44]).

This four-parameter distribution defines location, scale, and shape, and can be unimodal or bimodal. Its fitting performance has been compared to that of normal mixture models in simulation studies. Note that the beta distribution is a special case of the Dirichlet distribution (cf. Section 4.4.2).

#### 2.7.2. The Beta-Weibull Distribution

The beta-Weibull distribution, denoted by *BW*(*a*, *b*; *c*, *γ*), has the following pdf:(16)$$\begin{array}{c} f_{BW}\left(x\right)=\frac{1}{B(a,b)}\frac{c}{\gamma }{\left(\frac{x}{\gamma }\right)}^{c-1}{\left[1-\exp\left\{-{\left(\frac{x}{\gamma }\right)}^{c}\right\}\right]}^{a-1}\exp\left\{-b{\left(\frac{x}{\gamma }\right)}^{c}\right\},\;x>0. \end{array}$$
A system study on this distribution refer to Famege et al. [45] and Mdziniso [46]. The beta-Weibull distribution includes many useful distributions, including the Weibull distribution (*a* = *b* = 1), beta-exponential distribution (*c* = 1), Rayleigh distribution (*a* = *b* = 1, *c* = 2), two-parameter Burr-type distribution (*b* = 1, *c* = 2), etc.

The beta-Weibull distribution has many good properties.

(1)The limit of beta-Weibull density (Equation 16) is$$\begin{array}{c} \lim_{x\to \infty }f_{BW}\left(x\right)=\left\{\begin{array}{cc} \infty , & ac<1,\\ \frac{c\Gamma (a+b)}{\gamma \Gamma (a)\Gamma (b)}, & ac=1,\\ 0, & ac>1. \end{array}\right. \end{array}$$(2)Let *Y* ∼ *Be*(*a*, *b*), then the random variable $X=\gamma {\left[-\log(1-Y)\right]}^{1/c}$ follows *BW*(*a*, *b*, *c*, *γ*), where *c*, *γ* > 0. It provides a way to generate a random sample from the BW distribution.(3)Let *Y* follow a beta-exponential distribution with parameters (*a*, *b*, *θ*), then random variable *X* = *θ*(*Y*/*θ*)^1/*c*^ follows a beta-Weibull distribution.(4)The beta-Weibull distribution is unimodal. The mode is at *x*_0_ = 1 whenever *ac* < 1 or *ac* = 1 and **b** ≥ (*c* − 1)/2*c*. For other cases, *x*_0_ is the solution of the equation$$\begin{array}{c} bc-\left(c-1\right){\left(x_{0}/\gamma \right)}^{-c}=c\left(a-1\right){\left[\exp{\left(x_{0}/\gamma \right)}^{c}\right]}^{-1}. \end{array}$$(5)The *r*th moments of *BW*(*a*, *b*, *c*, *γ*) is given by$$\begin{array}{c} E\left[{\left(\frac{X}{\gamma }\right)}^{r}\right]=\frac{1}{B(a,b)}\int_{0}^{\infty }t^{r/c}{\left(1-e^{-t}\right)}^{a-1}e^{-bt}dt, \end{array}$$
by the transformation *t* = (*x*/*γ*)^*c*^.

## 3. Construction of Univariate Discrete Distributions and an Approximation

A discrete distribution is characterized by its support points and their probabilities: (17)$$\begin{array}{c} \begin{array}{c} \begin{array}{ccccc} X & x_{1} & x_{2} & \cdots & x_{k} \end{array}\\ \begin{array}{ccccc} p_{i} & p_{1} & p_{2} & \cdots & p_{k} \end{array} \end{array} \end{array}$$
where the random variable *X* has support $x_{1},\ldots ,x_{k}$ and $P\left(X=x_{i}\right)=p_{i},i=1,\ldots ,k.$

### 3.1. Classical Univariate Discrete Distributions

Standard discrete distribution model outcomes from fundamental random processes/trials:Bernoulli(*p*)/Binomial(*n*, *p*): The Bernoulli distribution, Bernoulli(*p*), models a single trial, and Binomial(*n*, *p*) counts successes in *n* independent trials with sample success probability *p*.Geometric(*p*): It models the number of Bernoulli trials needed to obtain the first success, representing the waiting time for a success. It is a special case of the negative binomial (with *r* = 1).Negative Binomial(*r*, *p*): As a generalization of the geometric distribution, it models the number of Bernoulli trials needed to achieve *r* successes. The negative binomial distribution presented here in its classic form (modeling the number of trials) is equivalent to the parameterization modeling the number of failures before the *r*-th success (with support y=0,1,2,…), which is prevalent in many modern software packages and generalized linear models.Poisson(*λ*): It models the number of events occurring in a fixed interval of time or space, given a constant average rate (*λ*) and independence between events. It can be derived as a limiting case of the binomial distribution when *n* → ∞ and *p* → 0 such that *np* → *λ*.Hypergeometric(*s*, *n*, *N*, *M*): It represents the number *s* of successes in *n* draws without replacement from a population with *N* size with *M* success items, where max(0, *n* − (*N* − *M*)) ≤ *s* ≤ min(*M*, *n*). This differs from the binomial distribution, where trials are independent (i.e., with replacement).

See Johnson and Kotz [47] for an extensive catalog. A guide to selecting common discrete probability distributions is illustrated in Figure 1, with probability mass functions summarized in Table 6.

Consider a random variable *X* following the discrete distribution defined in (Equation 17) and a continuous function *g*(·). The transformed variable *Y* = *g*(*X*) follows a distribution with the following structure: (18)$$\begin{array}{c} \begin{array}{ccccc} Y & g(x_{1}) & g(x_{2}) & \cdots & g(x_{k}) \end{array}\\ \begin{array}{ccccc} \;p_{i}\; & p_{1} & p_{2} & \cdots & p_{k} \end{array} \end{array}.$$
In other words, the law of *Y* is determined by mapping the support points of *X* via *g* while preserving the associated probabilities. This elegant property facilitates the analysis of transformed discrete variables. Li et al. [14] utilized this property to compute nearly representative points for the distribution of *g*(*X*).

### 3.2. Distributions Generated by Mixtures

A powerful method for constructing flexible probability models is through mixture distributions, where the parameter of one distribution is itself governed by another distribution. This approach naturally incorporates heterogeneity and over-dispersion, commonly observed in real-world data.

#### 3.2.1. Poisson Mixtures

When the rate parameter *λ* of a Poisson distribution is not fixed but follows a continuous distribution *G*(*λ*) with support on $R^{+}$, the resulting marginal distribution is a Poisson mixture. Its probability mass function is given by$$P\left(X=k\right)=\int_{0}^{\infty }\frac{e^{-\lambda }\lambda^{k}}{k!}dG\left(\lambda \right),\;k=0,1,2,\ldots$$
The most prominent example is the negative binomial distribution, which arises as a Poisson-gamma mixture. Specifically, ifX∣λ∼Poisson(λ)andλ∼Gamma(r,β=(1−p)/p),
then the marginal distribution of *X* is negative binomial with pmf:$$P\left(X=k\right)=\frac{\Gamma (r+k)}{k!\;\Gamma (r)}p^{r}{\left(1-p\right)}^{k},\;k=0,1,2,\ldots$$
This model is widely used for count data where the variance exceeds the mean.

#### 3.2.2. Binomial Mixtures

Similarly, allowing the success probability *p* of a binomial distribution to vary according to a continuous distribution *F*(*p*) on [0,1] leads to a binomial mixture. The marginal pmf isP(X=k)=∫01nkpk(1−p)n−kdF(p),k=0,1,…,n

The beta-binomial distribution is the canonical example, where *p* ∼ Beta(*α*, *β*). Its pmf isP(X=k)=nkB(k+α,n−k+β)B(α,β)
where B(·, ·) is the beta function. This distribution effectively models counts with extra-binomial variation.

Other notable mixtures include the Poisson-inverse Gaussian and the negative binomial-beta distributions [48].

### 3.3. Distributions Generated by Random Sums

Distributions generated by random sums, also known as compound distributions, provide a framework for modeling the aggregate outcome of a random number of events. Let *N* be a non-negative integer-valued random variable with pmf *p*_*N*_(*n*), and let {*X*_*i*_} be a sequence of *i*.*i*.*d*. random variables, independent of *N*. The random sum is defined as$$S=X_{1}+X_{2}+\cdots +X_{N},\;(\mathrm{with}\;\mathrm{the}\;\mathrm{convention}\;\mathrm{that}\;S=0\;\mathrm{if}\;N=0).$$
The distribution of *S* is called the compound distribution of *N* and *X*_*i*_. Its probability generating function (pgf) is given by *G*_*S*_(*t*) = *G*_*N*_(*G*_*X*_(*t*)), from which moments and probabilities can be derived.

Prominent examples include the following:Compound Poisson distribution: If *N* ∼ Poisson(*λ*), then *S* is compound Poisson. This is a highly flexible family used in insurance (to model total claim amounts) and queueing theory. Special cases include the Poisson-exponential and Poisson-gamma (which is equivalent to the negative binomial under a specific parameterization) distributions.Geometric sum: If *N* ∼ Geometric(*p*), and *X*_*i*_ ∼ Exp(*θ*), then *S* ∼ Exp(*θp*). This result is foundational in renewal theory.

The family of phase-type distributions can also be constructed using random sums of exponential random variables, demonstrating the versatility of this method for generating distributions with specific tail behaviors [49].

### 3.4. Approximation to Univariate Continuous Distributions by Representative Points

Consider a random variable *X* ∼ *F*(*x*), where *F*(*x*) is a univariate cumulative distribution function. The empirical distribution function *F*_*k*_(*x*) is defined as(19)$$\begin{array}{c} F_{k}\left(x\right)=\frac{1}{k}\sum_{i=1}^{k}I\left\{x_{i}\leq x\right\}, \end{array}$$
where *I*_*A*_ is the indicator function of *A*. The empirical distribution *F*_*k*_(*x*) is a discrete distribution with support points $x_{1},\ldots ,x_{k}$, each assigned an equal probability mass of 1/*k*. It serves as a consistent approximation of *F*(*x*), meaning *F*_*k*_(*x*) → *F*(*x*) in distribution as *k* → ∞. This distribution can be represented as(20)$$\begin{array}{c} \begin{array}{ccccc} Y_{MC} & x_{1} & x_{2} & \ldots & x_{k} \end{array}\\ \begin{array}{ccccc} \;p_{i}\; & \frac{1}{k} & \frac{1}{k} & \ldots & \frac{1}{k} \end{array} \end{array}.$$

Alternatively, one can construct other approximations using the so-called representative points (RPs) as support points. For a given distribution *F*(*x*) in R, several types of RPs exist, including

Monte Carlo RPs (MC-RPs): the empirical sample points.Quasi-Monte Carlo RPs (QMC-RPs): deterministic points based on low-discrepancy sequences.Mean square error RPs (MSE-RPs): points minimizing a mean squared error criterion.

For a univariate continuous distribution *F*(*x*), the QMC-RP approximation is given by the following distribution: (21)$$\begin{array}{c} \begin{array}{ccccc} Y_{QMC} & F^{-1}\left(\frac{1}{2k}\right) & F^{-1}\left(\frac{3}{2k}\right) & \ldots & F^{-1}\left(\frac{2k-1}{2k}\right) \end{array}\\ \begin{array}{ccccc} \;p_{i}\; & \;\frac{1}{k}\; & \;\frac{1}{k}\; & \ldots & \;\frac{1}{k}\; \end{array} \end{array}.$$
MSE-RPs and related approximation distribution is(22)$$\begin{array}{c} \begin{array}{ccccc} Y_{MSE} & \xi_{1} & \xi_{2} & \ldots & \xi_{k} \end{array}\\ \begin{array}{ccccc} \;p_{i}\; & p_{1} & p_{2} & \ldots & p_{k} \end{array} \end{array}.$$
where the support points $\left\{\xi_{1},\ldots ,\xi_{k}\right\}$ are determined by(23)$$\begin{array}{c} \mathrm{MSE}\left(Y\right)=\int_{-\infty }^{+\infty }\min_{i}{\left(x-\xi_{i}^{\left(k\right)}\right)}^{2}p\left(x\right)dx=\sum_{j=1}^{k}\int_{I_{j}}{\left(x-\xi_{i}^{\left(k\right)}\right)}^{2}p\left(x\right)dx, \end{array}$$
where(24)$$\begin{array}{cc} \; & I_{1}=\left(a_{1},a_{2}\right),\;I_{2}=\left(a_{2},a_{3}\right),\;\ldots ,\;I_{k}=\left(a_{k},a_{k+1}\right),\\ \; & a_{1}=-\infty ,\;a_{i}=\left(\xi_{i-1}^{\left(k\right)}+\xi_{i}^{\left(k\right)}\right)/2,\;i=2,\;\cdots ,\;k,\;a_{k+1}=\infty . \end{array}$$

Fang and Pan [50] and Fang et al. [8] provided a comprehensive review of the theory, generating algorithms, and applications of representative points in statistical inference.

#### 3.4.1. Difference Between Two Distributions

The Kullback–Leibler (KL) divergence, or relative entropy, of two probability distributions with probability density functions *p*(*x*) and *q*(*x*) on $X\subset R^{p}$ is defined as the expected logarithmic ratio of densities concerning *p*(*x*): (25)$$\begin{array}{c} D_{KL}\left(p\left(x\right)|q\left(x\right)\right)=\int_{X}\log\left(\frac{p(x)}{q(x)}\right)p\left(x\right)dx=E_{p(x)}\left[\log\left(\frac{p(x)}{q(x)}\right)\right]. \end{array}$$
The KL divergence is non-negative and is equal to 0 if and only if the two distributions are identical.

Now, consider the univariate case. For any discrete distribution, we can find a corresponding density estimator, denoted by $\hat{p}\left(x\right)$. Let *F*(*x*) be a cdf with pmf *p*(*x*), and let $\hat{F}\left(x\right)$ with pmf $\hat{p}\left(x\right)$ be an approximating distribution. Several criteria in the literature measure the distance between *F* and $\hat{F}$: $$\begin{array}{ccc} \mathrm{KL}\;\mathrm{divergence}(KL):\mathrm{KL}(p\|\hat{p}) & = & \sum_{x}\log\left(\frac{p(x)}{\hat{p}\left(x\right)}\right)p\left(x\right);\\ L^{2}\mathrm{-norm}\;\mathrm{of}\;\mathrm{pmfs}\left(L^{2}\mathrm{pdf}\right):∥p,\hat{p}∥ & = & {\left(\sum_{x}{\left[p\left(x\right)-\hat{p}\left(x\right)\right]}^{2}\;\right)}^{1/2};\\ L^{2}\mathrm{-norm}\;\mathrm{of}\;\mathrm{cdfs}\left(L^{2}\mathrm{cdf}\right):∥F,\hat{F}∥ & = & {\left(\int_{R}{\left[F\left(x\right)-\hat{F}\left(x\right)\right]}^{2}\;dx\right)}^{1/2}. \end{array}$$
Another metric for the distance between *F*(*x*) and $\hat{F}\left(x\right)$ is the absolute bias index (ABI), which evaluates the overall bias in parameter estimation. For a population distribution *F*(*x*; ***θ***) with parameter vector $\theta =(\theta_{1},\ldots ,\theta_{m})$ and its estimate $\hat{\theta }=\left({\hat{\theta }}_{1},\ldots ,{\hat{\theta }}_{m}\right)$ under $\hat{F}\left(x\right)$, the ABI is defined as follows:(26)$$\begin{array}{c} ABI=\frac{1}{m}\sum_{i=1}^{m}\left|\frac{\theta_{i}-{\hat{\theta }}_{i}}{\theta_{i}}\right|. \end{array}$$

#### 3.4.2. Discrete Analogues

Discrete analogs aim to create new discrete distributions that preserve one or more defining characteristics of an original continuous distribution. An approximation $\hat{F}\left(x\right)$ is considered successful if it maintains as much information from *F*(*x*) as possible, whether measured by the distance between the distributions, the similarity of their moments, or the properties of simulated samples.

As a continuous random variable *X* can be characterized by its pdf, moments, generating functions, or other features, the construction of a discrete analog *Y* (with pmf) from *X* (with density *f*(*x*)) focuses on matching these properties. The primary methods for this construction, summarized by [51], are classified as follows.
(1)Discrete analogs of the Pearson differential equation. This approach constructs discrete probability distributions by employing difference equations that mirror the structure of the Pearson differential equation, which defines the continuous Pearson family [52,53].(2)The probability mass function (pmf) of *Y* retains the form of the pdf of *X*, and the support of *Y* is determined from the full range of *X*. The pmf is defined as$$\begin{array}{c} P\left(Y=k\right)=\frac{f(k)}{\sum_{j=-\infty }^{\infty }f\left(j\right)},\;k=0,\pm 1,\pm 2,\ldots \;. \end{array}$$
This approach has been used to create discrete analogs of various continuous distributions, including the gamma, general Dirichlet, normal, lognormal, exponential, Laplace, generalized exponential, and gamma distributions [54,55,56].(3)The survival function (SF) of *Y* retains the form of the survival function of *X*, and the support of *Y* is determined from the full range of *X*. The discrete survival function is defined as *S*_*Y*_(*k*) = *P*(*Y* ≥ *k*), and its corresponding cdf is *F*_*Y*_(*k*) = *P*(*Y* ≤ *k*), related by *S*_*Y*_(*k*) = 1 − *F*_*Y*_(*k* − 1). This technique has generated numerous discrete distributions, such as the discrete exponential, Weibull, geometric Weibull, normal, Rayleigh, Maxwell, extended exponential, and Lindley distributions [57,58].(4)The hazard (failure) rate function of *Y* retains the form of the hazard rate function of *X*. This method preserves the hazard rate function. For a continuous random variable *X* with survival function *S*_*X*_(*x*) = *P*(*X* ≥ *x*) and hazard rate function *λ*_*X*_(*x*) = *f*_*X*_(*x*)/*S*_*X*_(*x*), the survival function of the discrete analog *Y* is given by$$\begin{array}{c} P\left(Y\geq k\right)=\prod_{i=1}^{k-1}\left(1-\lambda_{X}\left(i\right)\right),\;k=1,\ldots ,m, \end{array}$$
and its pmf is$$\begin{array}{c} P\left(Y=k\right)=\lambda_{X}\left(k\right)\prod_{i=1}^{k-1}\left(1-\lambda_{X}\left(i\right)\right). \end{array}$$
This approach has been used to propose new discrete distributions, such as the discrete Lomax, Weibull, and Rayleigh [59].(5)The moments of *Y* and *X* coincide up to a certain order. This method requires that *Y* and *X* share the same finite *r*^*th*^ moment for r=0,1,…,2N−1, and that their cdfs coincide at least at *M* + 1 points. The support of the discrete random variable *Y* is $\left\{y_{1},\ldots ,y_{MN}\right\}$. It has been used to define new discrete uniform, normal, gamma, and beta distributions [60].
There are more other construction methods, for instance, an extensive review on discrete distributions defined for integer data is provided in [61].

## 4. Multivariate Distributions

The commonly used discrete distributions can be extended to multivariate cases, including multinomial and multivariate negative binomial, hypergeometric, and Poisson distributions.

### 4.1. Multinomial Distribution

An extension of the binomial distribution is the so-called multinomial distribution. It is the fundamental model for counting events that fall into multiple categories, making it essential in fields like statistics, genetics, machine learning (e.g., topic modeling with naive Bayes), and survey analysis. Considering an experiment consists of *n* identical, independent trials, each trial results in one of *k* mutually exclusive and exhaustive categories with probabilities $p_{i},i=1,\ldots ,k;p_{1}+\ldots +p_{k}=1$. Let random vector $X=(X_{1},\ldots ,X_{k})$ follow a multinomial distribution $Multinomial(n;p_{1},\ldots ,p_{k})$. Its probability mass function is(27)$$\begin{array}{c} P\left(X_{1}=n_{1},\ldots ,X_{k}=n_{k}\right)=\frac{n!}{n_{1}!\cdots n_{k}!},n_{1},\ldots ,n_{k}\in N_{0},\;n_{1}+\ldots +n_{k}=n. \end{array}$$
The multinomial distribution has many nice properties. One is related to Poisson distribution, that lets $X_{1},\ldots ,X_{k}$ be independent and $X_{i}∼Poisson\left(\lambda_{i}\right),i=1,\ldots ,k$. Then the conditional distribution of $\left(X_{1},\ldots ,X_{k}|X_{1}+\cdots +X_{k}=n\right)$ follows $Multinomial(n;p_{1},\ldots ,p_{k})$ where $p_{i}=\lambda_{i}/\left(\lambda_{1}+\cdots +\lambda_{k}\right),i=1,\ldots ,k$. The multinomial distribution has been used in many statistical applications. One is in the Pearson’s chi-square test for goodness-of-fit, using the fact that the test statistic, calculated from observed multinomial counts and expected counts (*np*_*i*_), follows a chi-square distribution asymptotically (as *n* grows large).

### 4.2. Multivariate Hypergeometric Distribution

The multivariate hypergeometric distribution is the generalization of the hypergeometric distribution to more than two categories. It describes the probability of drawing specific numbers of items from each category without replacement. Considering a population with size *N*, of which $M_{1},\ldots ,M_{k}$ are the numbers of items in the *k* mutually exclusive and exhaustive categories with $\sum_{i=1}^{k}M_{i}=N$, suppose that the *n* random samples are sampling without replacement containing $n_{1},\ldots ,n_{k}$ items of each corresponding category. The joint distribution of $X=(X_{1},\ldots ,X_{k})$ has pmf(28)P(X1=n1,⋯,nk)=∏i=1kMiniNn,∑i=1kni=n,0≤ni≤Mi,i=1,⋯,k.
The distribution denoted as $Mult.\;Hypg.(n;M_{1},\ldots ,M_{k})$ is the so-called multivariate hypergeometric distribution. Without requiring the parameters in (Equation 28) to be non-negative, multivariate inverse hypergeometric, negative hypergeometric, and negative inverse hypergeometric distributions are then defined; see [62]. One property is that if ***X*** and ***Y*** are independent and each distributed as $Multinomial(n;p_{1},\ldots ,p_{k})$, then ***X***|(***X*** + ***Y*** = ***N***) follows *Mult*. *Hypg*. (*n*, ***N***).

### 4.3. Multivariate Poisson Distribution

If the parameters $p_{1},\cdots ,p_{k-1}$ in the multinomial distribution are allowed to tend to 0, this implies that *p*_*k*_ tends 1 as *n* → ∞, such that $np_{i}\to \lambda_{i},\;i=1,2,\cdots ,k-1$. Hence, the probability of obtaining the event $\cap_{i=1}^{k-1}\left(X_{i}=n_{i}\right)$ tends to(29)$$\begin{array}{c} P\left(X_{1}=n_{1},\cdots ,X_{k-1}=n_{k-1}\right)=\prod_{i=1}^{k-1}\left\{e^{-\lambda_{i}}\frac{\lambda_{i}^{n_{i}}}{n_{i}!}\right\}. \end{array}$$
This is the joint distribution of *k* − 1 mutually independent Poisson random variables with means $\lambda_{1},\cdots ,\lambda_{k-1}$. The distribution can also be obtained as a limiting form of negative multinomial distributions.

The simplest case, as derived from the multinomial limit, is one of independence, but more flexible distributions allowing for correlation are of greater practical interest. A more general and widely used approach to constructing a multivariate Poisson distribution with a covariance structure is to use a common mixture or latent variable model. A standard construction introduces a set of independent latent Poisson variables and defines the observed variables as their sums.

A classical bivariate Poisson distribution for random variables (*X*, *Y*) can be defined as follows:$$X=Z_{1}+Z_{0},\;Y=Z_{2}+Z_{0},$$
where *Z*_0_, *Z*_1_, *Z*_2_ are independent Poisson random variables with rates *λ*_0_, *λ*_1_, *λ*_2_ ≥ 0, respectively. The joint probability mass function of (*X*, *Y*) is given by$$P\left(X=x,Y=y\right)=\exp\left(-\left(\lambda_{1}+\lambda_{2}+\lambda_{0}\right)\right)\sum_{k=0}^{\min(x,y)}\frac{\lambda_{1}^{x-k}}{(x-k)!}\frac{\lambda_{2}^{y-k}}{(y-k)!}\frac{\lambda_{0}^{k}}{k!}.$$
The marginal distributions are *X* ∼ Poisson(*λ*_1_ + *λ*_0_) and *Y* ∼ Poisson(*λ*_2_ + *λ*_0_), and the covariance between *X* and *Y* is Cov(*X*, *Y*) = *λ*_0_ ≥ 0. This model can only account for non-negative correlation, which is often sufficient for many applications, such as modeling the number of insurance claims across different policy types or goals scored by two teams in a match.

This construction can be generalized to higher dimensions by introducing more common components, leading to a richer covariance structure [3,63,64]. These correlated multivariate Poisson models are essential tools in modern multivariate count data analysis.

### 4.4. Continuous Multivariate Distributions

Johnson et al. [4] provided a comprehensive compilation of many multivariate distributions. Johnson [7] offered an in-depth introduction to the generation of multivariate statistical simulations. Palmitesta and Provasi [65] reviewed a range of multivariate distributions and their computer generation methods. Their survey encompassed the Dirichlet distribution, the multivariate Johnson distribution system, multivariate beta and gamma distributions, the multivariate extreme value distribution, the inverse Gaussian multivariate distribution, the Cook–Johnson family of multivariate uniform distributions, and elliptically contoured distributions. They also examined techniques for generating random vectors with specified marginal distributions and correlation matrices. Fang, Kotz, and Ng [6] conducted a comprehensive study on symmetric multivariate distributions. Due to space limitations, this section focuses on selected multivariate distributions from the literature and meta-distributions constructed via copulas, see [66]. For more construction methods for multivariate distribution, see [67].

#### 4.4.1. Stochastic Representation

The traditional definition of the multivariate normal distribution relies on its density function. This approach, however, has several weaknesses:(i)It assumes that the explicit form of the density function is known.(ii)It requires the analytical evaluation of high-dimensional integrals to compute probabilities, moments, and marginal distributions.

The stochastic representation operator, denoted by $\overset{d}{=}$, is a powerful tool for constructing new distributions. If two random vectors ***X*** and ***Y*** have the same distribution, we write $X\overset{d}{=}Y$.


**Examples**


$U\overset{d}{=}1-U$, where *U* ∼ *U*(0, 1);

$X\overset{d}{=}-X$, where *X* ∼ *N*(0, *σ*^2^);

$X\overset{d}{=}\mu +\sigma Y$, where *X* ∼ *N*(*μ*, *σ*^2^) and *Y* ∼ *N*(0, 1).


**Properties**


(1)$X\overset{d}{=}Y$$\Rightarrow \left(f_{1}\left(X\right),\ldots ,f_{m}\left(X\right)\right)\overset{d}{=}\left(f_{1}\left(Y\right),\ldots ,f_{m}\left(Y\right)\right),$ where *f*_*j*_ are measurable functions.Examples are $X^{⊤}X\overset{d}{=}Y^{⊤}Y$ and $X/∥X∥\overset{d}{=}Y/∥Y∥.$(2)*X*, *Y*, and *Z* are random variables,$X\overset{d}{=}Y$ can imply $X+Z\overset{d}{=}Y+Z,$ if *Z* is independent of (*X*, *Y*).(3)If $X+Z\overset{d}{=}Y+Z$, and *Z* is independent of (*X*, *Y*), it is not necessary for $X\overset{d}{=}Y$.(4)If *Z* is independent of (*X*, *Y*), then
$X\overset{d}{=}Y$ implies $XZ\overset{d}{=}YZ$;$XZ\overset{d}{=}YZ$ implies $X\overset{d}{=}Y$ under some conditions.(5)$X\overset{d}{=}Y$ if and only if $X^{+}\overset{d}{=}Y^{+}$ and $X^{-}\overset{d}{=}Y^{-}$.

**Example** **1.***The multivariate normal distribution, denoted by X ∼ N_p_*(***μ***, **Σ**)*, has the following stochastic representation:*(30)$$\begin{array}{c} X\overset{d}{=}\mu +A^{⊤}Z, \end{array}$$*where *$\mu ={\left(\mu_{1},\ldots ,\mu_{p}\right)}^{⊤},A^{⊤}A=\Sigma$*, and *$Z={\left(Z_{1},\ldots ,Z_{p}\right)}^{⊤}$*with i*.*i*.*d*. *elements Z_i_ ∼ N*(0, 1)*. It is easy to find density function, moments, and marginal distributions of ***Z*** and then transfer these to **X**. For example, from the density function of **Z**, *$\frac{1}{{\left(\sqrt{2\pi }\right)}^{p}}e^{-\frac{1}{2}z^{⊤}z}$*, it is straightforward to find the density of **X** as*(31)$$\begin{array}{c} f\left(x\right)={\left(2\pi \right)}^{-p/2}{\left|\Sigma \right|}^{-1/2}\exp\left\{-\frac{1}{2}{\left(x-\mu \right)}^{⊤}\left(x-\mu \right)\right\}. \end{array}$$*As the mean and covariance of ***Z*** are, respectively, *0* and **I**_p_, one can immediately find E*(***X***) = ***μ** and Cov*(***X***) = Σ*, respectively. It is also easy to find that all the marginal and affine transformations of **X** are normally distributed. For more detailed discussion and applications, see [68].*

Many efforts to extend constructing methods from univariate to multivariate, for example, Pearson system, exponential family, and many others, can be found in [4].

#### 4.4.2. Dirichlet Distribution

Let *X*_1_, …, *X*_*m*−1_ and *X*_*m*_ be independent random variables, and *X*_*i*_ ∼ *Gamma*(*α*_*i*_), *i* = 1, …, *m*. Set$$\begin{array}{c} Y=X_{1}+\cdots +X_{m},\;\;Y_{j}=\frac{X_{j}}{Y},j=1,\ldots ,m. \end{array}$$
The distribution of ***Y*** = (*Y*_1_, …, *Y*_*m*_)^⊤^ is called the Dirichlet distribution with parameters $\alpha_{1},\ldots ,\alpha_{m}$ and is written as (*Y*_1_, …, *Y*_*m*_) ∼ *D*_*m*_(*α*_1_, …, *α*_*m*_).

The beta distribution *Be*(*α*, *β*) is a special case of Dirichlet distribution. As $Y_{1}+\ldots +Y_{m}=1$, the random vector $\left(Y_{1},\ldots ,Y_{m}\right)$ does not have a density on $R^{m}$. However, the density of the first *m* − 1 components (with respect to Lebesgue measure on the simplex) is given by$$\begin{array}{c} f\left(y_{1},\ldots ,y_{m-1}\right)=\frac{1}{B_{m}\left(\alpha \right)}\prod_{i=1}^{m}y_{i}^{\alpha_{i}-1},\;\sum_{i=1}^{m}y_{i}=1\;\;\;\;0\leq y_{i}\leq 1,i=1,\ldots ,m, \end{array}$$
where **α** = (*α*_1_, …, *α*_*m*_) and$$\begin{array}{c} B_{m}\left(\alpha_{1},\ldots ,\alpha_{m}\right)=\frac{\Gamma \left(\alpha_{1}\right)\cdots \Gamma \left(\alpha_{m}\right)}{\Gamma (\alpha_{1}+\cdots +\alpha_{m})}. \end{array}$$
Moreover, *Y* is independent of $\left\{Y_{j},j=1,\ldots ,m-1\right\}$.

There is a fundamental relationship between the Dirichlet distribution and the uniform distribution on the unit sphere *S*^*p*^ = {***x*** : ***x***^⊤^***x*** = 1} [68].

**Theorem** **2.***Suppose that **U**
*∼ *U*(*S^p^*) *in *$R^{p}$*. Then **U** has a stochastic representation*
(32)$$\begin{array}{c} U=\left[\begin{array}{c} U_{1}\\ \vdots \\ U_{m} \end{array}\right]\overset{d}{=}\left[\begin{array}{c} D_{1}U^{\left(1\right)}\\ \vdots \\ D_{m}U^{\left(m\right)} \end{array}\right], \end{array}$$
*where $U^{\left(j\right)}\in R^{t_{j}},\;t_{1}+\cdots +t_{m}=p$, and*
(i)***U***^(*j*)^*∼ U*(*S^t_j_^*)*, the uniform distribution on *$S^{t_{j}},j=1,\cdots ,m$*;*(ii)*$U^{\left(1\right)},\cdots ,U^{\left(m\right)}$ and $\left(D_{1},\cdots ,D_{m}\right)$ are mutually independent;*(iii)*$D_{j}>0,j=1,\cdots ,m$, and $\left(D_{1}^{2},\cdots ,D_{m}^{2}\right)∼D_{m}\left(t_{1}/2,\cdots ,t_{m}/2\right)$*.

Based on the Dirichlet distribution, the *multivariate Liouville distribution* is defined on $R_{+}^{m}$ by the stochastic representation $X\overset{d}{=}RY$, where *R* > 0 is a random variable independent of **Y** and $Y∼D_{m}\left(\alpha_{1},\ldots ,\alpha_{m}\right)$. For a comprehensive study of the Liouville distribution, see Chapter 6 of [6].

#### 4.4.3. Spherical and Elliptical Distributions

There are a number of equivalent ways to define the spherical distribution. Let ***X*** be a *p* × 1 random vector. The following statements are equivalent [6]:(i)$X\overset{d}{=}PX$ for each ***P*** ∈ *O*(*p*), where *O*(*p*) is the set of orthogonal matrices of order *p*;(ii)The characteristic function (c.f.) of ***X*** is of the form *ψ*(***t***^⊤^***t***) for some *ψ* ∈ Ψ_*p*_ where(33)$$\begin{array}{c} \Psi_{p}=\left\{\psi \left(\cdot \right):\psi \left(t_{1}^{2}+\ldots +t_{p}^{2}\right)\mathrm{is}\;a\;\mathrm{p-dimensional}\;\mathrm{c.f.}\right\}; \end{array}$$(iii)***X*** has a stochastic representation $X\overset{d}{=}RU^{\left(p\right)}$, where random variable *R* ≥ 0 is independent of ***U***^(*p*)^, ***U***^(*p*)^ is uniformly distributed on the unit sphere *S*^*p*^ and does not have a density;(iv)For any $a\in S^{p}\;\mathrm{and}\;b\in S^{p}$, we have $a^{⊤}X\overset{d}{=}b^{⊤}X$.

**Theorem** **3.**
*Let $X\overset{d}{=}RU^{\left(p\right)}$ and P(X=0)=0. Then*

$$\left\|X\right\|\overset{d}{=}R,\;\;X/\left\|X\right\|\overset{d}{=}U^{\left(p\right)},$$

*and they are independent.*


**Example** **2.***The theorem has two immediate applications. First, it provides a method to generate a random sample from the uniform distribution on the sphere, U*(*S^p^*)*: simply sample **X** from *$N_{p}\left(0,I_{p}\right)$*and normalize it to obtain *$\frac{X}{\left\|X\right\|}$*. Second, it allows the mixed moments of **U***^(*p*)^*to be easily calculated from the known mixed moments of the standard normal vector **X** and the moments of its radius R *= ||***X***||*. A detailed discussion can be found in Section 3.1.2 of [6].*

**Theorem** **4.***Let **X**
*∼ *S_p_*(*ψ*) *denote a spherically symmetric distribution in *$R^{p}$*, characterized by its characteristic function ψ*(***t***^⊤^***t***) *as (Equation 33). It is not necessary that **X** has a density. The random vector *$X\overset{d}{=}RU^{\left(p\right)}$
*has a density which must be of the form g*(***x***^⊤^***x***)*, if and only if the radial variable R has a density, denoted by f*(*r*)*. The relationship between g and f is given by*
$$f\left(r\right)=\frac{2\pi^{p/2}}{\Gamma (p/2)}r^{p-1}g\left(r^{2}\right),$$
*where g*(·) *is called the density generator. We write **X**
*∼ *S_p_*(*g*).*The necessary condition that **X**
*∼ *S_p_*(*ψ*) *has a density is*
$$\int_{0}^{\infty }y^{p/2-1}g\left(y\right)dy<\infty$$
*as*
$$\int g\left(x^{⊤}x\right)dx=\frac{\pi^{p/2}}{\Gamma (p/2)}\int_{0}^{\infty }y^{p/2-1}g\left(y\right)dy=1.$$

Table 7 gives many useful spherical distributions from [6].


**Elliptical Distribution**


A random vector **X** has an elliptical distribution, denoted as **X** ∼ *EC*_*p*_(*μ*, **Σ**), with parameters ***μ*** and **Σ** = **A**^⊤^**A**, if it can be expressed as(34)$$\begin{array}{c} X\overset{d}{=}\mu +A^{⊤}Y\overset{d}{=}\mu +RA^{⊤}U^{\left(p\right)}, \end{array}$$
where **Y** ∼ *S*_*p*_(*ψ*) is spherical. This relationship generalizes the way a normal variable *X* ∼ *N*(*μ*, *σ*^2^) is constructed from a standard normal *Z* ∼ *N*(0, 1). Consequently, any subclass of spherical distributions in Table 7 defines a corresponding subclass of elliptical distributions.

#### 4.4.4. Approximation to Multivariate Continuous Distributions by Representative Points

For multivariate continuous distributions, the QMC-RPs and MSE-RPs can be defined in a similar way to the univariate case.Suppose that a random vector ***X*** ∼ *F*(·) in $R^{n}$ has a density function *f*(·) with a finite mean vector and covariance matrix. A set of points $\Xi =\{\xi_{j},j=1,\ldots ,k\}$ is called mean squared error representative points if it minimizes the mean squared error, $\mathrm{MSE}\left(\Xi \right)=\int_{R^{n}}\min_{j=1,\ldots ,k}{∥x-\xi_{j}∥}^{2}f\left(x\right)\;dx,$ where ∥ · ∥ denotes the *L*^2^-norm. For information regarding the construction of approximation distributions of a continuous multivariate distribution by their QMC-RPs, one can refer to [8,69,70]. For information regarding the construction of approximation distributions of a continuous multivariate distribution by their MSE-RPs, one can refer to [8,70,71,72,73]. Due to the complexity of computation in high-dimensional multivariate distribution, the related research works about MSE-RPs of this kind of distribution are limited. Representative points also have many applications for discrete data, especially for discrete massive data. In Chapter 9 of [8], some cases where the sample size is hugely larger than the number of variables have been considered, and some model-based and model-free subsampling methods are included.

### 4.5. Basic Statistics Under the Multivariate Population

Let ***X*** follow a multivariate continuous distribution *F*(***x***; ***θ***) where ***θ*** represents parameters. A random sample $x_{1},\ldots ,x_{n}$ is drawn from this population. The distributions of basic statistics, such as the mean vector, the covariance matrix, and various test statistics, are fundamental to statistical inference and its applications. For example, when the population distribution is multivariate normal, *N*_*p*_(***μ***, **Σ**), the standard unbiased estimators for ***μ*** and **Σ** are sample mean and the sample covariance, given, respectively, by(35)$$\begin{array}{c} \hat{\mu }=\bar{x}=\frac{1}{n}\sum_{i=1}^{n}x_{i},S=\frac{1}{n-1}\sum_{i=1}^{n}{\left(x_{i}-\bar{x}\right)}^{⊤}\left(x_{i}-\bar{x}\right). \end{array}$$
It is known that $\bar{x}∼N_{p}\left(\mu ,\frac{1}{n-1}\Sigma \right)$. Other key related distributions are described below.

Wishart distribution: The Wishart distribution, denoted by *W*_*p*_(*n*, **Σ**), is the multivariate generalization of the chi-square distribution. It is defined by $W=\sum_{i=1}^{n}Z_{i}Z_{i}^{⊤}$ and $Z_{i}∼N_{p}\left(0,\Sigma \right)$ are *i*.*i*.*d*. The sample covariance matrix **S** follows Wishart distribution: (*n* − 1)**S** ∼ *W*_*p*_(*n* − 1, **Σ**). This distribution was developed by John Wishart [74], who, in his seminal 1928 paper “The Generalised Product Moment Distribution in samples from a normal multivariate population”, published in Biometrika, derived the distribution of the matrix of sums of squares and cross-products. A crucial result for normally distributed data is that the sample mean $\bar{x}$ and the sample covariance matrix **S** are independent, generalizing the corresponding univariate property.

Distributions of the eigenvalues of **S**: Unlike the sample covariance matrix itself, the distributions of its individual eigenvalues are complex and do not follow simple, universal forms like the normal or Wishart distributions.

Hotelling’s T-squared distribution: It is the multivariate generalization of the Student’s *t*-distribution, used primarily for hypothesis testing on the mean vector. For a random vector ***X*** ∼ *N*_*p*_(***μ***, **Σ**) independent of **S** ∼ *W*_*p*_(*n*, **Σ**), the Hotelling’s *T*^2^ statistic is denoted as(36)$$\begin{array}{c} T^{2}=nX^{⊤}S^{-1}X. \end{array}$$
The resulting *T*^2^ statistic, despite being derived from multivariate data, is a scalar. Therefore, its sampling distribution, denoted by *T*^2^(*p*, *n*, ***μ***), is a univariate distribution. When μ=0, the distribution is central and denoted by *T*^2^(*p*, *n*). The *T*^2^ statistic is related to the *F*-distribution through the following identity: $\frac{n-p+1}{np}T^{2}\left(p,n\right)∼F\left(p,n-p+1\right).$

Wilks’ lambda distribution: This distribution generalizes the beta distribution and is used in multivariate analysis of variance and likelihood ratio tests. Let *A*_1_ ∼ *W*_*p*_(*m*, **Σ**) and *A*_2_ ∼ *W*_*p*_(*n*, **Σ**) be independent. The Wilks’ lambda statistic, defined as $\Lambda =\frac{A_{2}}{A_{1}+A_{2}}$, is a scalar. Thus, its sampling distribution, denoted by Λ(*p*, *m*, *n*), is a univariate distribution. Wilks’ lambda has close relationships with the *T*^2^ distribution and simplifies to an *F* or beta distribution when *p* = 1.

When the population **X** follows an elliptical distribution (a broader class than the normal), deriving the distributions of the aforementioned statistics becomes significantly more challenging. A comprehensive discussion of this topic can be found in [68].

#### Multivariate *L*_1_-Norm Symmetric Distribution

The elliptical distribution can be regarded as an extension of the multivariate normal distribution. Similarly, the exponential distribution can be generalized to a multivariate setting. Let $X_{1},\ldots ,X_{p}$ be *i*.*i*.*d*. random variables from the standard exponential distribution, *Exp*(1) (see Table 4). Consider the random vector $X={\left(X_{1},\ldots ,X_{p}\right)}^{⊤}$ and its *L*_1_-norm defined by ${\left\|X\right\|}_{1}=|X_{1}\left|+\ldots +\right|X_{p}|$. Let $U=\frac{X}{{\left\|X\right\|}_{1}}$. Fang and Fang [75] defined a class of multivariate distributions via the following stochastic representation:(37)$$\begin{array}{c} X\overset{d}{=}RU, \end{array}$$
where the radial variable *R* is independent of the directional vector **U**. This class of distributions was studied systematically in [75,76] and in Chapter 5 of [6].

### 4.6. Definition by Distribution, Density or Survival Functions

#### 4.6.1. Meta-Skew Symmetric Distributions

A traditional approach to defining a probability distribution is through its distribution function, density function, or survival function. For instance, the multivariate normal distribution in Anderson’s book [77] is defined by its density function. A general univariate form of an asymmetric distribution, known as the skew-symmetric family, can be represented as a function that combines a cumulative distribution function and a probability density function; see [78,79].

**Definition** **2.***A random variable X, whose density function has the form*(38)$$\begin{array}{c} p(x)=2f(x)G(w(x)), \end{array}$$*is called skew symmetric, where f*(*x*) *is a density function centrally symmetric about 0, w*(*x*) *is an odd real-valued function such that w*(−*x*) = −*w*(*x*)*, and G*(*w*(·)) *is a cumulative distribution function on *R*such that *$g=G^{\prime }$ *is an even density function, providing G is differentiable.*

#### 4.6.2. Skew-Normal Distribution

The multivariate skew-normal (MSN) distribution, introduced in Table 3, is a fundamental and powerful generalization of both the multivariate normal and the univariate skew-normal distributions. It provides a flexible framework for modeling multivariate data that exhibits asymmetry (skewness).

A *p*-dimensional random vector ***Y*** follows a multivariate skew-normal distribution with location vector ***μ***, positive definite scale matrix **Σ**, and skewness vector ***α*** if its probability density function is given by$$\begin{array}{c} f\left(y;\mu ,\Sigma ,\alpha \right)=2\phi_{p}\left(y-\mu ;\Sigma \right)\Phi \left(\alpha^{⊤}\omega^{-1}\left(y-\mu \right)\right), \end{array}$$
where *ϕ*_*p*_(*z*; **Σ**) is the pdf of *p*-variate normal distribution with mean 0 and covariance matrix **Σ**; Φ(·) is the cdf of *N*(0, 1); **ω** is the diagonal matrix formed from the standard deviations of **Σ** (the square roots of the diagonal elements), and the vector **α** is the shape or skewness parameter vector. It denotes *Y* ∼ *SN*_*p*_(***μ***, **Σ**, **α**).

The MSN distribution admits an intuitive stochastic representation. Consider a (*p* + 1)-dimensional normal random vector ${\left(X_{0},X_{1}^{⊤}\right)}^{⊤}$,(39)$$\begin{array}{c} \left[\begin{array}{c} X_{0}\\ X_{1} \end{array}\right]∼N_{1+p}\left[\left(\begin{array}{c} 0\\ 0 \end{array}\right),\left(\begin{array}{cc} 1 & \delta^{⊤}\\ \delta & \Sigma \end{array}\right)\right]. \end{array}$$
Here, *X*_0_ is a scalar and ***X***_1_ is a *p*-vector. The vector ***δ*** controls the correlation between *X*_0_ and each component of ***X***_1_. Then$$\begin{array}{c} Y\overset{d}{=}\mu +\omega \left(X_{1}|X_{0}>0\right)∼SN_{p}\left(\mu ,\Sigma ,\alpha \right), \end{array}$$
where the skewness parameter ***α*** relates to ***δ*** through the following equation: $$\begin{array}{c} \alpha =\frac{1}{\sqrt{1-\delta^{⊤}\Sigma^{-1}\delta }}\Sigma^{-1}\delta . \end{array}$$
For comprehensive details on the theory and applications of the multivariate skew-normal distribution, see [80,81].

#### 4.6.3. Skew-Elliptical Distributions

Skew-elliptical distributions form a broad family that generalizes symmetric elliptical distributions by introducing skewness (asymmetry). This extension moves beyond the familiar world of symmetric, bell-shaped curves into the asymmetric realm, providing a principled and mathematically sound framework for modeling a wider range of real-world data patterns. Among these, the skew-*t* distribution is often a particularly useful starting point for applied work due to its flexibility in handling both skewness and heavy tails.

The general form of the pdf for a skew-elliptical distribution is(40)$$\begin{array}{c} f\left(z\right)=2f_{0}\left(z\right)\Pi \left(z\right), \end{array}$$
where *f*_0_(*z*) is the pdf of a symmetric elliptical distribution (the “parent”), and Π(*z*) is a skewing function that maps onto the interval [0, 1] and satisfies Π(*z*) + Π(−*z*) = 1. This function controls the direction and amount of skewness. The skew-elliptical family is vast, but several members are particularly important: skew-normal, skew-*t*, skew-Cauchy, and skew-Laplace distributions. The skew-elliptical distributions are used anywhere that data deviates from symmetry and normality, such as in finance, environmental science, biology, psychometrics, and social sciences. For a comprehensive study, refer to [82].

#### 4.6.4. Definition by Survival Function

The survival function of a random variable *X* ∼ *F*(*x*) is defined as $\bar{F}\left(x\right)=P\left(X>x\right)$. This function is particularly useful in the analysis of biometrical data. Within this domain, the exponential and Weibull distributions play an important role.

A foundational model for dependent failures is the Marshall–Olkin multivariate exponential distribution [83]. Its strength lies in an intuitive shock-based construction and analytical tractability. The model assumes that shocks to the system components arrive according to three independent Poisson processes with rates *λ*_1_, *λ*_2_, and *λ*_12_, corresponding to shocks that fatally affect only component 1, only component 2, or both components simultaneously. The arrival times of these shocks, *Z*_1_, *Z*_2_, and *Z*_12_, are, therefore, independent exponential random variables with parameters *λ*_1_, *λ*_2_, and *λ*_12_, respectively.

We set *X* = min{*Z*_1_, *Z*_12_} and *Y* = min{*Z*_2_, *Z*_12_}. The joint survival function of the random vector (*X*, *Y*) is defined as $\bar{H}\left(x,y\right)=P\left(X>x,Y>y\right)$. For any *x*, *y* ≥ 0 the survival function is given by$$\begin{array}{ccc} \bar{H}\left(x,y\right) & = & P\left(Z_{1}>x\right)P\left(Z_{2}>y\right)P\left(Z_{12}>\max\left(x,y\right)\right)\\ & = & \exp[-\lambda_{1}x-\lambda_{2}y-\lambda_{12}\max\left(x,y\right)]. \end{array}$$
Its marginal survival functions are$$\begin{array}{c} \bar{F}\left(x\right)=\exp\left(-\left(\lambda_{1}+\lambda_{12}\right)x\right),\mathrm{and}\;\bar{G}\left(y\right)=\exp\left(-\left(\lambda_{2}+\lambda_{12}\right)y\right). \end{array}$$
Hence, *X* and *Y* are exponential random variables with parameters *λ*_1_ + *λ*_12_ and *λ*_2_ + *λ*_12_, respectively. This elegant construction can be extended to higher dimensions.

### 4.7. Meta-Distributions via Copula Techniques

Various statistical distributions play a key role in modeling real-world data. The rapid development of data science has increased the demand for a wider variety of multivariate distributions. A common challenge is to construct a multivariate distribution from given one-dimensional marginal distributions while incorporating a known dependence structure. Such constructions arise not only from statistical theory but also from practical problems, particularly in risk and decision analysis for complex multivariate systems. The copula is a powerful technique designed for this purpose.

Copulas are mathematical functions that fully capture the dependence structure among random variables, offering great flexibility in building multivariate stochastic models. Since their introduction in the 1950s, copulas have gained significant popularity in several applied fields such as finance, insurance, and reliability theory. Nowadays, they are a well-recognized tool for market and credit modeling, risk aggregation, portfolio selection, and more.

The following definition is based on probability theory. For a more general treatment, see [66,84]. As most research works and applications are interested in the copulas with respect to continuous distributions, and it becomes complicated for discrete distributions, this section only covers the continuous case.

**Definition** **3.***If *$C(z_{1},\ldots ,z_{p})$*is the cumulative distribution function on the unit hypercube *[0, 1]*^p^ with uniform U*(*0*, *1*) *margins, then C is called a copula.*

The following Lemma demonstrates the existence of the copula function.

**Lemma** **1.***Let *$H(x_{1},\ldots ,x_{p})$*be a continuous distribution with p continuous margins F_i_ on the domain D_i_ for *i=1,…,p. *Then there exists a unique copula C such that*(41)$$\begin{array}{c} C\left(z_{1},\ldots ,z_{p}\right)=H\left(F_{1}^{-1}\left(z_{1}\right),\ldots ,F_{p}^{-1}\left(z_{p}\right)\right). \end{array}$$

It is well known that for a given set of *p* marginal distributions ${F_{i}\left(x_{i}\right)}_{i=1}^{p}$, there exist infinitely many *p*-dimensional distributions with these margins. If the dependence structure of the joint distribution is known, one may find a unique or a small family of such joint distributions $H(x_{1},\ldots ,x_{p})$. The copula technique is a fundamental tool for this purpose.

Suppose we have *p* random variables *X*_*i*_ ∼ *F*_*i*_(*x*_*i*_) for i=1,…,p, and we wish to construct a multivariate distribution *H* with these margins and a given dependence structure defined by a copula $C(z_{1},\ldots ,z_{p})$. This method is called the copula technique. The resulting distribution is termed a *meta-distribution* and is denoted by $F(C;F_{1},\ldots ,F_{p})$.

**Theorem** **5.***Sklar’s Theorem (Multivariate Case) Let **X** be a continuous random vector with joint distribution *$H(x_{1},\ldots ,x_{p})$* and continuous marginal distributions F_i_ on domains D_i_ for *i=1,…,p. *Then there exists a unique copula C such that*(42)$$\begin{array}{c} H\left(x_{1},\ldots ,x_{p}\right)=C\left(F_{1}\left(x_{1}\right),\ldots ,F_{p}\left(x_{p}\right)\right). \end{array}$$*Conversely, if C is a copula and *$F_{1},\ldots ,F_{p}$* are univariate distribution functions, then the function H defined by (Equation 42) is a p-dimensional distribution function with margins *$F_{1},\ldots ,F_{p}$.
*If the margins F_i_ are continuous, then C is unique. The joint density function of **X** derived from (Equation 42) is given by*

(43)
$$\begin{array}{c} h\left(x_{1},\ldots ,x_{p}\right)=c\left(F_{1}\left(x_{1}\right),\ldots ,F_{p}\left(x_{p}\right)\right)\prod_{i=1}^{p}f_{i}\left(x_{i}\right), \end{array}$$

*where f_i_ is the probability density function of X_i_, and c(·) is the copula density, i.e., the p-th mixed partial derivative of C.*


#### 4.7.1. Some Examples of Copula

Many copula examples exist for lower dimensions. This subsection presents selected examples from Nelsen [66] to provide a tutorial for beginners to comprehend the ideas of copula. Examples 3 and 4 have different joint distributions but have the same copula.

**Example** **3.**
*Adapted from Nelsen [66]*
*Consider the joint distribution*$$\begin{array}{c} H\left(x,y\right)=\frac{\left(x+1\right)\left(e^{y}-1\right)}{x+2e^{y}-1},\;\left(x,y\right)\in \left[-1,1\right]\times \left[0,\infty \right). \end{array}$$*Its marginal distributions are X *∼ *U*(−1, 1) *with cdf *$F\left(x\right)=\frac{x+1}{2}$
*and Y *∼ *Exp*(1)*, the standard exponential distribution with cdf G*(*y*) = 1 − *e*^−*y*^*, and copula*
$$\begin{array}{c} C\left(u,v\right)=\frac{uv}{u+v-uv}. \end{array}$$
*We now demonstrate how to derive the copula C from H, and, conversely, how to reconstruct H from C, F, G.*
*Deriving C from H:*
*Let u *= *F*(*x*) *and v *= *G*(*y*)*, which implies *$u=\frac{x+1}{2}$*and v *= *1 *− *e*^−*y*^
*or *$e^{y}=\frac{1}{1-v}$*.*
$$\begin{array}{ccc} C(u,v) & & =H\left(F^{-1}\left(u\right),G^{-1}\left(v\right)\right)=\frac{\left(F^{-1}\left(u\right)+1\right)\left(e^{G^{-1}\left(v\right)}-1\right)}{F^{-1}\left(u\right)+2e^{G^{-1}\left(v\right)}-1}\\ & & =\frac{\left[\left(2u-1\right)+1\right]\left[e^{-\ln(1-v)}-1\right]}{2u-1+2e^{-\ln(1-v)}-1}\\ & & =\frac{2u(\frac{1}{1-v}-1)}{2u-1+2\frac{1}{1-v}-1}=\frac{2u[1-(1-v)]}{\left(2u-1)(1-v)+2-(1-v\right)}\\ & & =\frac{uv}{u+v-uv}. \end{array}$$
*Deriving H from *{*C*, *F*, *G*}*:*
*Note that *

$e^{y}=\frac{1}{1-v}$

*; we have*

$$\begin{array}{ccc} H(x,y) & & =C(u,v)\\ & & =\frac{\frac{x+1}{2}\left(1-e^{-y}\right)}{\frac{x+1}{2}+1-e^{-y}-\frac{x+1}{2}\left(1-e^{-y}\right)}\\ & & =\frac{\left(x+1\right)\left(1-e^{-y}\right)}{x+1+2\left(1-e^{-y}\right)-\left(x+1\right)\left(1-e^{-y}\right)}\\ & & =\frac{\left(x+1\right)\left(e^{y}-1\right)}{\left(x+1\right)+2e^{y}-2}=\frac{\left(x+1\right)\left(e^{y}-1\right)}{x+2e^{y}-1}. \end{array}$$



**Example** **4.**
*Gumbel’s bivariate logistic distribution [85].*

$$\begin{array}{c} H\left(x,y\right)={\left(1+e^{-x}+e^{-y}\right)}^{-1},x,y\geq 0. \end{array}$$

*Its two margins are*

$$\begin{array}{c} F\left(x\right)={\left(1+e^{-x}\right)}^{-1},\;G\left(y\right)={\left(1+e^{-y}\right)}^{-1}. \end{array}$$

*Its copula is*

$$\begin{array}{c} C\left(u,v\right)=\frac{uv}{u+v-uv}\;, \end{array}$$

*which is identical to the copula in Example 3, despite the marginal distributions being quite different.*


For multivariate copulas, higher-dimensional margins are defined as per the following example.

**Example** **5.**
*Consider the following three-dimensional distribution H:*

$$\begin{array}{c} H\left(x,y,z\right)=\frac{\left(x+1\right)\left(e^{z}-1\right)\sin\left(z\right)}{x+2e^{y}-1},\;on\left[-1,1\right]\times \left[0,\infty \right]\times \left[0,\frac{\pi }{2}\right]. \end{array}$$

*Its three one-dimensional marginal distributions are*

$$\begin{array}{c} H_{1}\left(x\right)=\frac{x+1}{2},H_{2}\left(y\right)=1-e^{-y},H_{3}\left(z\right)=\sin\left(z\right). \end{array}$$

*The three two-dimensional marginal distributions are*

$$\begin{array}{ccc} & & H_{12}\left(x,y\right)=\frac{\left(x+1\right)\left(e^{y}-1\right)}{x+2e^{y}-1},\\ & & H_{23}\left(y,z\right)=\left(1-e^{-y}\right)\sin\left(z\right),\\ & & H_{13}\left(x,z\right)=\frac{1}{2}\left(x+1\right)\sin\left(z\right). \end{array}$$



**Example** **6.***Fang et al. [86] provided a comprehensive study of the copula defined for α *> 0 *and β *∈ [0, 1]*:*
(44)$$\begin{array}{c} C\left(u,v;\alpha ,\beta \right)=\frac{uv}{{\left[1-\beta \left(1-u^{1/\alpha }\right)\left(1-v^{1/\alpha }\right)\right]}^{\alpha }}. \end{array}$$
*For different values of α and β, we have the following:*
*1.* *β *= 0; *U and V are independent.**2.* *β *= 1; *it reduces to the bivariate Clayton distribution [87].**3.* *α *= 1; *it reduces to the Ali–Mikhail–Haq copula [88].*
*Its multivariate version is given by*
(45)$$\begin{array}{c} C\left(u_{1},\ldots ,u_{n};\alpha ,\beta \right)=\frac{\prod_{i=1}^{n}u_{i}}{{\left[1-\beta \prod_{i=1}^{n}\left(1-u_{i}^{1/\alpha }\right)\right]}^{\alpha }}. \end{array}$$

These examples demonstrate that the copula technique provides a powerful method for constructing numerous new distributions useful for statistical modeling.

#### 4.7.2. Gaussian Copula and Meta-Gaussian Distribution

When *H*(·) is a multivariate normal distribution, the associated copula is called a Gaussian copula. When *H*(·) is an elliptical distribution, the corresponding copula is known as an elliptical copula. This family includes the Gaussian copula, the *t*-copula, and the Kotz-type copula, among others; for a comprehensive overview, see [89].

From the definition of a copula, the Gaussian copula for *N*_*p*_(***μ***, **Σ**) is identical to that of $Z=\left(Z_{1},\ldots ,Z_{p}\right)∼N_{p}\left(0,R\right)$, where ***R*** is the correlation matrix of **Z**. Clearly, all marginal distributions of **Z** are standard normal *N*(0, 1), yet its components have correlation matrix ***R*** = (*r*_*ij*_). Denote the cdf of **Z** by $G_{\mathrm{normal}}\left(z_{1},\ldots ,z_{p}|R\right)$ and define a function *C* as follows:(46)$$\begin{array}{c} C_{normal}\left(u_{1},\ldots ,u_{p}\right)=G_{normal}\left(\Phi^{-1}\left(u_{1}\right),\ldots ,\Phi^{-1}\left(u_{p}\right)\right),\;{\left(u_{1},\ldots ,u_{p}\right)}^{⊤}\in {\left[0,1\right]}^{p}, \end{array}$$
where Φ(·) is the cdf of *N*(0, 1). This function *C* is the Gaussian copula. Kelly and Krzysztofowicz [90] studied the meta-Gaussian (or meta-normal) distribution. A more detailed analysis can be found in Section 2.6, “Multivariate Gaussian/Normal”, of [91].

**Definition** **4.**
*The meta-Gaussian or meta-normal distribution with a set of given marginal cdfs $F_{1},\ldots ,F_{p}$ is defined by its cdf*

(47)
$$\begin{array}{c} F_{X_{1},\ldots ,X_{p}}\left(x_{1},\ldots ,x_{p}\right)=G_{Z_{1},\ldots ,Z_{p}}\left(\Phi^{-1}\left(F_{1}\left(x_{1}\right)\right),\ldots ,\Phi^{-1}\left(F_{p}\left(x_{p}\right)\right)\right). \end{array}$$

*We denote this distribution as $X∼MN_{p}\left(0,R;F_{1},\ldots ,F_{p}\right)$.*


The pdf of $MN_{p}\left(0,R;F_{1},\ldots ,F_{p}\right)$ is$$\begin{array}{c} h\left(x_{1},\ldots ,x_{p}\right)=c_{normal}\left(F_{1}\left(x_{1}\right),\ldots ,F_{p}\left(x_{p}\right)\right)\prod_{i=1}^{p}f_{i}\left(x_{i}\right), \end{array}$$
where *c*_*normal*_ is the density of the Gaussian copula and *f*_*i*_ is the pdf corresponding to *F*_*i*_. In the case where the correlation matrix has all off-diagonal elements equal to *ρ*, Bouy’e [92] provided illustrative plots for various values of *ρ*.

Let $X∼MN_{2}\left(0,R;F_{1},F_{2}\right)$, where the correlation matrix with Corr(*X*_1_, *X*_2_) = *ρ*. Then the normal copula is given by$$\begin{array}{c} C_{normal}\left(u,v\right)=\frac{1}{2\pi \sqrt{1-\rho^{2}}}\int_{-\infty }^{\Phi^{-1}\left(u\right)}\int_{-\infty }^{\Phi^{-1}\left(v\right)}\exp\left\{-\frac{s^{2}-2\rho st+t^{2}}{2(1-\rho^{2})}\right\}dsdt, \end{array}$$
where 0 ≤ *u*, *v* ≤ 1.

The correlation coefficient has been used in multivariate analysis for a long time. However, the copula and its related meta-distributions might have different correlation coefficients due to different marginal distributions. Therefore, the so-called “measure of concordance” is considered in the literature. It requests that the copula and its meta-distributions have the same values of the measure of concordance. The Kendall’s *τ* and the Spearman’s *rho* are the most popular ones used in the applications. Their definitions can refer to [66]. Spearman’s *ρ* for the binormal copula is given by $\frac{6}{\pi }\arcsin\left(\frac{\rho }{2}\right)$ (see [93]). This result does not extend to other elliptical distributions, which are introduced in the next section.

The meta-normal distribution $X∼MN_{2}\left(0,R;F_{1},F_{2}\right)$ is given by(48)$$\begin{array}{ccc} F(x_{1},x_{2}) & & =C_{normal}\left(F_{1}\left(x_{1}\right),F_{2}\left(x_{2}\right)\right)\\ & & =\frac{1}{2\pi \sqrt{1-\rho^{2}}}\int_{-\infty }^{\Phi^{-1}\left(F_{1}\left(x_{1}\right)\right)}\int_{-\infty }^{\Phi^{-1}\left(F_{2}\left(x_{2}\right)\right)}\exp\left\{-\frac{s^{2}-2\rho st+t^{2}}{2(1-\rho^{2})}\right\}dsdt. \end{array}$$

Mai and Scherer [94], in Example 1.7, discussed the Gaussian copula and meta-Gaussian distributions for the general case ***X*** ∼ *N*_*p*_(***μ***, **Σ**). In this case, the marginal distributions of *X*_*i*_ (i=1,…,p) are not necessarily identical if the variances Var(*X*_*i*_) differ. Now, consider the two-dimensional normal distribution $N_{2}\left(0,R\right)$ with correlation matrix ***R*** and correlation coefficient *ρ*. Let *ξ*_*i*_ = Φ^−1^(*u*_*i*_) for *i* = 1, 2. The density of the Gaussian copula, as given in Chapter 7 of [95], is(49)$$\begin{array}{c} c_{normal}\left(\xi \right)=c_{normal}\left(\xi_{1},\xi_{2}\right)={\left|R\right|}^{-1/2}\exp\left(-\frac{1}{2}\xi^{⊤}\left(R^{-1}-I\right)\xi \right). \end{array}$$
This expression can be factorized into two independent components$$\begin{array}{c} c_{normal}\left(\xi \right)=\frac{1}{2\pi \sqrt{1-\rho^{2}}}\exp\left(-\frac{1}{2}\xi^{⊤}R^{-1}\xi \right)[2\pi \cdot \exp\left(\frac{1}{2}\left(\xi_{1}^{2}+\xi_{2}^{2}\right))\right]. \end{array}$$
The first factor is the density of $N_{2}\left(0,R\right)$, while the second factor is the product of the densities of two independent standard normal variables (up to a constant),



$$\sqrt{2\pi }\exp\left(\frac{1}{2}\xi_{1}^{2}\right)\mathrm{and}\sqrt{2\pi }\exp\left(\frac{1}{2}\xi_{2}^{2}\right).$$



Recently, He et al. [96] derived the covariance matrix of the Gaussian copula and related properties.

#### 4.7.3. Marshall–Olkin Copula

The Marshall–Olkin multivariate exponential distribution was introduced in Section 4.6.4. We now derive its corresponding survival copula, denoted by ${\hat{C}}_{MO}\left(u,v\right)$. We begin by expressing the joint survival function $\bar{H}\left(x,y\right)$ in terms of the marginal survival functions $u=\bar{F}\left(x\right)$ and $v=\bar{G}\left(y\right)$, recalling the relationship $\bar{H}\left(x,y\right)=\hat{C}\left(\bar{F}\left(x\right),\bar{G}\left(y\right)\right)$, where $\hat{C}$ is the survival copula. Using the identity max(*x*, *y*) = *x* + *y* − min(*x*, *y*), we can rewrite the survival function as follows:$$\begin{array}{ccc} \bar{H}\left(x,y\right) & = & \hat{C}\left(\bar{F}\left(x\right),\bar{G}\left(y\right)\right)\\ & = & \exp[-\left(\lambda_{1}+\lambda_{12}\right)x-\left(\lambda_{2}+\lambda_{12}\right)y+\lambda_{12}\min\left(x,y\right)]\\ & = & \bar{F}\left(x\right)\bar{G}\left(y\right)\min\left\{\exp\left(\lambda_{12}x\right),\exp\left(\lambda_{12}y\right)\right\}. \end{array}$$
As $u^{\frac{-\lambda_{12}}{\lambda_{1}+\lambda_{12}}}=exp\left(\lambda_{12}x\right)$ and $v^{\frac{-\lambda_{12}}{\lambda_{2}+\lambda_{12}}}=exp\left(\lambda_{12}y\right)$, let $\alpha =\frac{\lambda_{12}}{\lambda_{1}+\lambda_{12}}$ and $\beta =\frac{\lambda_{12}}{\lambda_{2}+\lambda_{12}}$. Obviously, 0 < *α*, *β* < 1, and let$$\begin{array}{ccc} u & = & \bar{F}\left(x\right)=e^{-(\lambda_{1}+\lambda_{12})x}=e^{-\lambda_{12}x/\alpha },\;u^{-\alpha }=e^{\lambda_{12}x},\\ v & = & \bar{G}\left(y\right)=e^{-(\lambda_{2}+\lambda_{12})y}=e^{-\lambda_{12}y/\beta },\;v^{-\beta }=e^{\lambda_{12}y}. \end{array}$$
The survival copula for the Marshall–Olkin distribution, known as the Marshall–Olkin copula, is given by(50)$$\begin{array}{ccc} {\hat{C}}_{MO}\left(u,v\right) & & =uv\min\left\{u^{\frac{-\lambda_{12}}{\lambda_{1}+\lambda_{12}}},v^{\frac{-\lambda_{12}}{\lambda_{2}+\lambda_{12}}}\right\}=\min\left\{vu^{1-\frac{\lambda_{12}}{\lambda_{1}+\lambda_{12}}},uv^{1-\frac{\lambda_{12}}{\lambda_{2}+\lambda_{12}}}\right\}\\ & & =uv\min\left(u^{-\alpha },v^{-\beta }\right)=\min\left(u^{1-\alpha }v,uv^{1-\beta }\right). \end{array}$$
Therefore, the survival copula for the Marshall–Olkin bivariate exponential distribution forms a two-parameter family of copulas given by(51)$$\begin{array}{c} C_{\alpha ,\beta }\left(u,v\right)=\left\{\begin{array}{cc} u^{1-\alpha }v, & \mathrm{if}u^{\alpha }\geq v^{\beta },\\ uv^{1-\beta }, & \mathrm{if}u^{\alpha }\leq v^{\beta }. \end{array}\right. \end{array}$$

The application of copulas and meta-distributions is a cornerstone of modern quantitative finance, risk management, and many other fields that deal with multivariate dependence. They provide a powerful toolkit for modeling complex relationships beyond simple correlation.

## 5. Conclusions

The development of statistical distributions, driven by the need to model real data, is a cornerstone of statistical science. With a vast and well-established body of literature, a complete review of construction methods is beyond our scope. This article, instead, focuses on recent advancements in the field, particularly in simulation techniques and computational algorithms. Our aim is to offer readers an expanded perspective on the evolving landscape of statistical distributions.

## Figures and Tables

**Figure 1 entropy-27-01188-f001:**
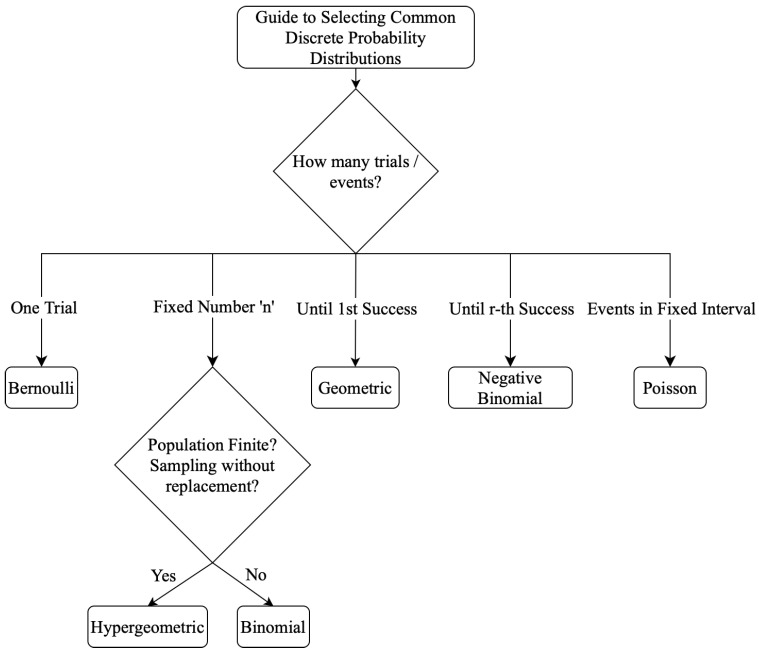
A guide to selecting common discrete probability distributions.

**Table 1 entropy-27-01188-t001:** Classification of Pearson distributions.

Type	Condition	Distribution	PDF Example
0	*a* = 0, *b* = 0	Normal distribution	$f\left(x\right)\propto e^{-{\left(x-\mu \right)}^{2}/2\sigma^{2}}$
I	*b*^2^ − 4*ac* > 0	Beta distribution	*f*(*x*) ∝ (1 + *x*)^*r*_1_^(1 − *x*)^*r*_2_^, |*x*| ≤ 1
II	*b*^2^ − 4*ac* = 0, *λ* = − *b*/2*a*	Symmetric beta	$f\left(x\right)\propto {\left(1-x^{2}\right)}^{r},\left|x\right|\leq 1$
III	*a* = 0	Gamma distribution	*f*(*x*) ∝ *x*^*k*−1^*e*^−*x*/*θ*^. *x* > 0
IV	*b*^2^ − 4*ac* < 0	Skewed heavy-tailed	Complex form
V	*c* = 0	Inverse gamma	*f*(*x*) ∝ *x*^*k*−1^*e*^−*θ*/*x*^, *x* > 0
VI	*b*^2^ − 4*ac* > 0, λ≠root	Beta prime, *F*-distribution	*f*(*x*) ∝ *x*^*r*_1_^(1 + *x*)^−(*r*_1_+*r*_2_)^
VII	*b* = 0	Student’s *t*-distribution	$f\left(x\right)\propto {\left(1+x^{2}/\nu \right)}^{-(\nu +1)/2}$

**Table 2 entropy-27-01188-t002:** Distributions of the order statistics.

Distribution	Density f(x)
The general k-th order statistic ^1^	(nk−1)[F(x)]k−1[1−F(x)]n−kf(x)
Minimum (*X*_(1)_)	*n*[1 − *F*(*x*)]^*n*−1^*f*(*x*)
Maximum (*X*_(*n*)_)	*n*[*F*(*x*)]^*n*−1^*f*(*x*)

^1^ Note: The pdf for the median exists in a closed form only when *n* is odd (*n* = 2*m* + 1), given by $p_{\mathrm{median}}\left(x\right)=\frac{n!}{{\left(m!\right)}^{2}}{\left[F\left(x\right)\right]}^{m}{\left[1-F\left(x\right)\right]}^{m}f\left(x\right)$. For even *n*, the median is defined as $\frac{1}{2}\left(X_{\left(n/2\right)}+X_{\left(n/2+1\right)}\right)$ and its distribution is derived from the joint distribution of these two order statistics.

**Table 6 entropy-27-01188-t006:** Famous discrete distributions.

Distribution	Support Points	Probability Mass Function $p_{i}\left(x\right)$
Bernoulli(*p*)	0, 1	*p*_0_ = 1 − *p*, *p*_1_ = *p*
Binomial(*n*, *p*)	0,1,2,…,n	pi=nipi(1−p)n−i,i=0,1,2,…,n
Geometric(*p*)	1,2,…	$p_{i}=p{\left(1-p\right)}^{i-1},i=1,2,\ldots$ (Number of trials until first success)
Negative Binomial(*r*, *p*)	r,r+1,…	pi=i−1r−1pr(1−p)i−r,i=r,r+1,… (Number of trials until *r*-th success)
	0,1,2,…	pi=i+r−1ipr(1−p)i,i=0,1,2,… (Number of failures before *r*-th success)
Poisson(*λ*)	0,1,2…	$p_{i}=e^{-\lambda }\frac{\lambda^{i}}{i!},i=0,1,2,\ldots$
Hypergeometric(*s*, *n*, *N*, *M*)	max (0, *n* − (*N* − *M*)), ⋯, min (*M*, *n*)	pi=MiN−Mn−i/Nn

Note on alternative derivations and parameterizations: The distributions listed above can often be derived through multiple probabilistic mechanisms, which leads to different (but equivalent) parameterizations and interpretations. These alternative derivations highlight the interconnectedness of discrete probability distributions and provide flexibility in choosing the most appropriate model for a given data-generating process.

**Table 7 entropy-27-01188-t007:** Some subclasses of *p*-dimensional spherical distributions.

Type	Density Function $g(x^{⊤}x)$ or c.f. $\psi (t^{⊤}t)$
Multinormal	$g\left(x^{⊤}x\right)=c\;\exp\left(-\frac{1}{2}x^{⊤}x\right)$
Kotz-type	$g\left(x^{⊤}x\right)=c{\left(x^{⊤}x\right)}^{N-1}\exp\left[-r{\left(x^{⊤}x\right)}^{s}\right],\;r,s>0,\;2N+p>2$
Pearson type VII	$g\left(x^{⊤}x\right)=c{\left(l+x^{⊤}x/s\right)}^{-N},\;N>p/2,s>0$
Multivariate *t*	$g\left(x^{⊤}x\right)=c{\left(l+x^{⊤}x/v\right)}^{-(p+v)/2},\;v>0$
Multivariate Cauchy	$g\left(x^{⊤}x\right)=c{\left(l+x^{⊤}x\right)}^{-(p+1)/2}$
Pearson Type II	$g\left(x^{⊤}x\right)=c{\left(l-x^{⊤}x\right)}^{q},\;q>0$ for ***x***^⊤^***x*** < 1
Logistics	$g\left(x^{⊤}x\right)=c\;\exp\left(-x^{⊤}x\right)/{\left[1+\exp\left(-x^{⊤}x\right)\right]}^{2}$
Scale mixture	$g\left(x^{⊤}x\right)=c\int_{0}^{\infty }t^{-p/2}\exp\left(-x^{⊤}x/2t\right)dG\left(t\right),G\left(t\right)\;a\;\mathrm{cdf}.$

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
