# Peer review of "A Review: Construction of Statistical Distributions"

_entropy, 2025, doi:10.3390/e27121188_

Round 1
Reviewer 1 Report
Comments and Suggestions for Authors
Honestly speaking, although I appreciated the efforts the authors put in this review of listing almost all the probabilistic distributions useful for hypotheses testing in stats-oriented models, I could not understand the effective added value of this work. I mean that the manuscript is overloaded of the formal maths, and that can be fine, even though it is possible to find all the details in many statistical books, but that's that. Even when the authors propose improvements concerning copulas and meta-distributions enhancement they stop at the formal maths. What is lacking are simulations or attempts of setting up some simulations to demonstrate such improvement or suggestion and thus the extent to which those distributions can be helpful. My view is that this paper is a mere academic exercise, but nothing more.
Reviewer 2 Report
Comments and Suggestions for Authors
see attachment

Reviewer 3 Report
Comments and Suggestions for Authors
A lot of interesting distributions of random processes could be reviewed as well. May be in a future?

Round 2
Reviewer 1 Report
Comments and Suggestions for Authors
Let me say that I found the authors' reply to my first review very useless. The authors pointed out what I had already stated last time. To my view, every theoretical improvements in any field the world over need to be corroborated by means of simulations of those theoretical issues. Especially related to stastistical advances. If you do not prove any simulations, even lab-simulations (I mean numerical) the result is like having a good car with an excellent engine without knowing if it starts or not simly because the car-maker did not want to start it up, but merely leaving to others if his/her car works or not.
Current research needs of new theory-driven stats improvements that work while solving practical situation of estimations, but these improvements need to be simulated, otherwise these improvements remain like a mere accademic excercise.
Reviewer 2 Report
Comments and Suggestions for Authors
I am fine with the added material and the changes made.
I am happy that most of the comments are used to imporve the manuscipt.